# Development of a Safe and Highly Efficient Inactivated Vaccine Candidate against Lumpy Skin Disease Virus

**DOI:** 10.3390/vaccines9010004

**Published:** 2020-12-23

**Authors:** Janika Wolff, Tom Moritz, Kore Schlottau, Donata Hoffmann, Martin Beer, Bernd Hoffmann

**Affiliations:** Institute of Diagnostic Virology, Friedrich-Loeffler-Institut, Federal Research Institute for Animal Health, Südufer 10, D-17493 Greifswald-Insel Riems, Germany; janika.wolff@fli.de (J.W.); Tom.Moritz@gmx.de (T.M.); kore.schlottau@fli.de (K.S.); donata.hoffmann@fli.de (D.H.); martin.beer@fli.de (M.B.)

**Keywords:** capripox, inactivated vaccine, LSDV, lumpy skin disease, vaccine, adjuvants

## Abstract

Capripox virus (CaPV)-induced diseases (lumpy skin disease, sheeppox, goatpox) are described as the most serious pox diseases of livestock animals, and therefore are listed as notifiable diseases under guidelines of the World Organisation for Animal Health (OIE). Until now, only live-attenuated vaccines are commercially available for the control of CaPV. Due to numerous potential problems after vaccination (e.g., loss of the disease-free status of the respective country, the possibility of vaccine virus shedding and transmission as well as the risk of recombination with field strains during natural outbreaks), the use of these vaccines must be considered carefully and is not recommended in CaPV-free countries. Therefore, innocuous and efficacious inactivated vaccines against CaPV would provide a great tool for control of these diseases. Unfortunately, most inactivated Capripox vaccines were reported as insufficient and protection seemed to be only short-lived. Nevertheless, a few studies dealing with inactivated vaccines against CaPV are published, giving evidence for good clinical protection against CaPV-infections. In our studies, a low molecular weight copolymer-adjuvanted vaccine formulation was able to induce sterile immunity in the respective animals after severe challenge infection. Our findings strongly support the possibility of useful inactivated vaccines against CaPV-infections, and indicate a marked impact of the chosen adjuvant for the level of protection.

## 1. Introduction

Lumpy skin disease virus (LSDV), sheeppox virus (SPPV) and goatpox virus (GTPV) are the three species of the genus *Capripoxvirus* within the *Poxviridae* family [1]. Capripox virus-induced diseases are described as globally the most serious pox diseases of livestock animals [2,3]. Clinical infections range from sub-clinical through mild to acute [3]. Transmission of LSDV occurs mainly mechanically via blood-feeding insects [4,5,6,7]. In addition, mechanical transmission via ticks [8,9,10,11] or via common use of drinking troughs of infected an naïve cattle [12] as well as seminal transmission [13] have been reported. Until now it is unknown where the virus resides during non-vector seasons [14]. After an incubation period of 4 to 14 days after experimental infections [15,16,17,18], 6 to 27 days after experimental infection using possible vector species (*Stomoxys Calcitrans*, *Haematopota spp*.) [19] and 1–4 weeks in natural outbreaks [12,18,20], an initial period of fever occurs [16,21]. Additionally, affected animals suffer from excessive salivation and nasal discharge [20,22], emaciation [22,23], enlarged lymph nodes [21,22] as well as characteristic skin lesions that either may occur in sporadic or generalized forms [16,17,21,22]. Direct losses appear due to drop in milk production [21,24], mass loss [21], decreased growth rate [23], as well as temporary or permanent infertility [14,21,23,24]. Moreover, expensive control and eradication measures are necessary in case of an outbreak [20]. The economic damage caused by capripox virus infections has categorized them as OIE-listed diseases [20,25]. Eradicating the diseases is difficult and time consuming regardless of different options described by OIE guidelines [20].

For control of Capripox viruses, a combination of movement restrictions and quarantine, stamping-out procedures and ring-vaccinations are recommended [2,20,21,26]. However, vaccination is described as the only effective way to control lumpy skin disease (LSD) [27,28] due to the vector-borne spread of the virus [4,5,6,7]. Until now, only modified live vaccines are commercially available [14], which were derived from field isolates [29,30] and were attenuated by multiple passages in cell culture [31] or in the chorioallantoic membrane of embryonated chicken eggs [32]. For vaccination of cattle against LSDV, homologous attenuated LSDV strains [27] are available, as well as potentially smuggled unlabeled LSDV vaccines [33]. In addition, in some areas, heterologous vaccines on basis of different SPPV strains [33,34,35] and GTPV strains [34,35] are used with an increased dose in cattle (10× the dose recommended for small ruminants). In some cases, no adverse effects could be observed after vaccination [36,37]. Nevertheless, various side-effects are reported for the live-attenuated vaccines: from rise in body temperature [33,38] through drop in milk production [33,39] and local reactions at the injection site [23,39,40] to sporadic or generalized LSD-like lesions [33,40,41] and clinical symptoms similar to clinical LSD [33,36]. However, in a recent study dealing with adverse effects after vaccination with live-attenuated vaccine against LSDV neither significant drop in milk production for 30 days post immunization nor increased mortality for 60 days post vaccination could be observed [42]. Furthermore, vaccine virus genome was detected in several studies in different matrices such as skin (nodes) [40], hair samples obtained from skin nodules [39], nasal swabs [40], milk [40], buffy coat [39] and whole blood samples [33,39,43]. Moreover, vaccine virus isolation was successful from skin nodules [40]. Therefore, spread of vaccine virus via blood-feeding insects cannot be fully excluded [39]. This is supported by the findings of Kononov et al., who examined different field outbreaks in Russia in 2017 and thereby detected four outbreaks which were caused by vaccine-like LSDV strains [44]. However, protection after vaccination with homologous vaccine strains is highly effective against any severe clinical course or death [28,30] and immunity is probably life-long [30]. Moreover, the GTPV vaccine “Caprivac” was also able to protect cattle from challenge infection to a great extent [37]. However, not all cattle are completely protected after vaccination with homologous [20,37] or heterologous [33,36] modified-live vaccines. Furthermore, several possible vaccine failures are reported, which were reviewed by Hunter and Wallace (2001) as well as Tuppurainen and Oura (2012). Additionally, mechanical transmission during a vaccination campaign via needles and diluents contaminated with field strains as well as failures during inoculation or incorrect vaccine doses are possible reasons for vaccine breakdowns [20]. Furthermore, over-attenuation of certain vaccine virus strains can also not be excluded [37].

Due to all above mentioned reasons, inactivated vaccines against Capripox viruses would be a desirable alternative, since they are non-replicating and thereby in most cases safe [45,46]. Although there are publications claiming only a short duration of protection and less immunogenicity after vaccination with inactivated Capripox virus vaccines [47], some studies revealed promising results regarding inactivated vaccines against Capripox viruses. More precisely, complete protection against SPPV challenge infection could be achieved after vaccination with inactivated SPPV strains already around 40 years ago [48], and protection of sheep against virulent challenge infection to 6 month post vaccination is reported [49]. In 2016, Boumart et al. published the results of their detailed study comparing the efficacy of a live-attenuated and an inactivated SPPV vaccine. After challenge infection, sheep vaccinated with inactivated SPPV did not show any clinical signs typical for SPPV infections, only increased body temperature for two days and a hypersensitivity reaction at the inoculation site of the challenge infection could be observed. Moreover, the protection index of the inactivated SPPV vaccine was comparable to that of the modified-live vaccine [45]. Similar results were obtained during different trials using homologous inactivated vaccines against GTPV [50,51]. However, sheep and goats vaccinated with an inactivated prototype vaccine were only partially protected against heterologous SPPV challenge infection [52]. To our knowledge, only a single publication exists dealing with an inactivated vaccine against LSDV, which was published recently by Hamdi et al. reporting the efficacy of an inactivated vaccine against LSDV in comparison to a commonly used live-attenuated LSDV vaccine. For the inactivated prototype vaccine, an attenuated LSDV-“Neethling” strain was inactivated using BEI, and an oily emulsion with Montanide adjuvant (SEPPIC) was intramuscularly given to cattle. In addition to protection against challenge infection, safety and immunogenicity of the inactivated vaccine were analyzed [46].

In our study, we performed two different animal trials testing the potential and efficacy of an inactivated LSDV vaccine and the impact of two different adjuvants on safety and efficiency of an inactivated vaccine. During the first trial, a proof-of-concept study, we vaccinated cattle with the BEI-inactivated LSDV-“Neethling vaccine” strain. Cattle immunized with the live-attenuated vaccine “Herbivac LS” served as a vaccine control group. Since we achieved good clinical protection against challenge infection with the virulent LSDV-“Macedonia2016“ field strain, a second animal trial was performed. Here, we used another LSDV isolate (LSDV-“Serbia“ field strain) for antigen preparation. Furthermore, we compared two different adjuvant preparations regarding side-effects and their impact on vaccine efficacy after challenge infection. Overall, cattle vaccinated with the inactivated LSDV-“Serbia“ field strain were completely protected against the severe challenge infection with virulent LSDV-“Macedonia2016“ field strain. Moreover, no shedding of viral genome could be detected at any time point during the animal trial. However, we saw clear differences in safety and efficacy of both tested adjuvants. In summary, findings of Hamdi et al. (2020) could be validated by our results and we confirmed that an inactivated vaccine against LSDV provides a safe and highly efficient tool for preventive vaccination campaigns when using the right adjuvant.

## 2. Materials and Methods

### 2.1. General

#### 2.1.1. Animals

For the proof-of-concept animal trial (trial 1) eighteen and for the vaccine-efficacy study (trial 2) twenty-four 4- to 6-month-old Holstein-Friesian cattle were divided into groups of six cattle each. The animals were housed in the facilities of the Friedrich-Loeffler-Institut Insel Riems under biosafety level 4 (animal; BSL3^AG^) conditions. All respective animal protocols were reviewed by a state ethics commission and have been approved by the competent authority (State Office for Agriculture, Food Safety and Fisheries of Mecklenburg-Vorpommern, Rostock, Germany, ref. LALLF M-V/TSD/7221.3-1-061/16 (FLI 13/16) from 20 October 2016).

#### 2.1.2. Inactivation of Virus Using Binary Ethylenimine

For inactivation of virus, binary ethylenimine (BEI) was used as it is a standard procedure for inactivation of viruses for commercial inactivated vaccines, e.g., foot-and-mouth disease virus vaccine [53,54]. Therefore, 200 mM sodium thiosulphate and 0.1 M 2-Bromoethylamine hydrobromide (BEA) in 200 mM sodium hydroxide were prepared and the BEA-NaOH mixture was incubated for 1 h at 37 °C for cyclisation reaction. Afterwards, 27 mL of antigen were mixed carefully with 3 mL of cyclic BEA (BEI) and incubated for 24 h at 28 °C. Inactivation reaction was stopped by adding 3 mL 200 mM sodium thiosulphate and careful mixing. Successful inactivation was validated by three passages on Madin-Darby bovine kidney (MDBK) cells (FLI cell culture collection number CCLV-RIE0261) and pan-Capripox real-time qPCR analyses (see Section 2.1.5. Molecular diagnostics).

#### 2.1.3. Clinical Score after Challenge Infection

Body temperature was measured daily from 4 days post vaccination (dpv) until 28 days post challenge (dpc). Here, fever was defined as body temperature >39.5 °C. Additionally, clinical reaction was determined based on a modified and enhanced clinical reaction score system of Carn & Kitching [16,55]. Clinical scoring was performed in a non-blinded manner.

#### 2.1.4. Sample Collection

Samples were taken at certain time points of both animal trials (Figure 1 and Figure 2).

During the proof-of-concept study (Figure 1), EDTA-blood and serum samples for analyses of cell-associated and cell-free viremia, respectively, as well as nasal, oral and ocular swabs for analyses of viral shedding were taken at days 0 and 21 of the animal trial in the inactivated LSDV-vaccinated group, and at Day 35 of the animal trial in all three groups. Additionally, samples were taken from Herbivac-vaccinated cattle at days 16, 19, 21 and 23 of the animal trials (representing 2, 5, 7, 9 dpv). After challenge infection, the samples were taken at 2, 5, 7, 9, 12, 14, 17, 21, 28 dpc for molecular analyses. Serum samples of 7, 14, 21, and 28 dpc were also analyzed serologically (Figure 1).

EDTA-blood, serum samples and nasal swabs were taken during the vaccine-efficacy study at -1 dpv as well as 3, 5, 7, 10, 12, 14, 17, 21, 28 dpc. Additionally, serum samples were taken at 21 days after primary immunization (in the following defined as dpv) and 42 dpv (day of challenge infection) (Figure 2). Skin samples from nodules were taken at different time points during the trial, depending on their appearance. After section, cervical lymph nodes and of the control group also mediastinal and mesenterial lymph nodes were analyzed regarding their viral genome load.

#### 2.1.5. Molecular Diagnostics

Lymph node samples as well as skin samples were homogenized in serum-free medium (FLI cell culture medium ZB5d, containing MEM (H), MEM (E) + non-essential amino acids) with antibiotics using the TissueLyser II tissue homogenizer (QIAGEN, Hilden, Germany). For DNA extraction of all samples of both animal trials, the KingFisher Flex System (Thermo Scientific, Darmstadt, Germany) was used with the NucleoMag Vet kit (Macherey-Nagel, Düren, Germany) according to the manufacturer’s instructions. An internal control DNA (IC-2 DNA) was added during the DNA extraction process for control of successful DNA extraction and inhibition free amplification [56]. For analyses of viral genome loads in all samples, the already described pan-Capripox real-time qPCR [57] with a modified probe [58] was performed using the PerfeCTa qPCR ToughMix (Quanta BioSciences, Gaithersburg, MD, USA).

#### 2.1.6. Virus Isolation

100 µL of homogenized skin samples were incubated on MDBK cells with approx. 90% confluence with less than 1 mL of serum-free medium in a T25 cell culture flask for 2 h at 37 °C. Subsequently, inoculated cells were washed 3× with cell culture medium. Then, 10 mL of cell culture medium with 10% fetal calf serum (FCS) (FLI cell culture medium ZB5) and antibiotics were added and the inoculated cells were incubated at 37 °C. After 7 days, cells were analyzed regarding a cytopathic effect (CPE) under a light microscope.

#### 2.1.7. Serological Examination

Serological analyses were performed using two different methods: A Double Antigen (DA) enzyme-linked immunosorbent assay (ELISA) and the serum neutralization test (SNT).

The ID Screen Capripox Double Antigen ELISA (ID.vet, Montepellier, France) was performed following the manufacturer’s instructions.

Serum samples for the LSDV-specific SNT were heat-inactivated for at least 30 min at 56 °C. Afterwards, log_2_ dilution series in serum-free medium were prepared in triplicates in 96 well plate format, starting from 1:10. A total of 50 µL of LSDV-“Neethling vaccine“ strain with a titer of 10^3.3^ cell culture infectious dose_50_ (CCID_50_)/mL were added to each well and the mixtures of diluted serum samples and virus were incubated for 2 h at 37 °C and 5% CO_2_. In the following, MDBK cells with a concentration of approx. 30,000 cells/100 µL were added. After incubation for 7 d at 37 °C and 5% CO_2_, development of CPE was analyzed using a light microscope (Nikon Eclipse TS-100). Determination of the SNT titer was performed with the Spearman–Kärber method [59,60].

### 2.2. Proof-of-Concept Study

#### 2.2.1. Vaccine Preparation

MDBK cells were seeded in 3× T175 non-vented cell culture flasks using 50 mL of cell culture medium with 10% FCS (ZB5) and incubated for 24 h at 37 °C. Then, the medium was removed and 1 mL of an LSDV-“Neethling vaccine“ strain (internal lab number V/100) with a titer of 10^6.5^ CCID_50_/mL in 10 mL ZB5 were used to infect the cell monolayer. After 2 h incubation at 37 °C, additional 40 mL of ZB5 were added. Infected cells were harvested at 4 days post infection (dpi) by freezing–thawing of the cells as a strong CPE was observed. The harvested virus was centrifuged for 5 min at 4000 rpm, the supernatant was removed, and the pellet was re-suspended in Tris-EDTA (TE) buffer pH 8.0 for concentration of virus. Finally, a titer of 10^7.5^ CCID_50_/mL was defined for the pooled virus stock. After standard inactivation with BEI, the inactivated antigen was stored at −80 °C until usage. 

LSDV-“Neethling vaccine” strain was chosen as it is closely related to the live-attenuated vaccine strains used for commercial vaccines. In this first proof-of-concept study, impact of different virus isolates should be excluded to gain insight whether an inactivated vaccine against LSDV is efficient or not.

For the final proof-of-concept vaccine, the Adjuvant A (10% Polygen, MVP Adjuvants^®^, Omaha, USA) was added to the inactivated LSDV. Each animal of the respective group received 2 mL of the vaccine preparation for both the primary and secondary vaccination.

#### 2.2.2. Experimental Design and Challenge Infection

Cattle of Group 1A were used as challenge control group. Vaccination with the commercially available Herbivac LS was used as vaccine positive control. Therefore, cattle of Group 1B were vaccinated subcutaneously (s.c.) with 2 mL Herbivac LS (LSDV-“Neethling vaccine“ with a titer of >10^2.5^ CCID_50_/mL, Batch/Lot: LSF1-0001; Exp: 04/2017) following the manufacturer’s instructions [61] at Day 14 of the animal trial. Cattle of Group 1C should be vaccinated with the same volume as cattle of Group B and therefore received 2 mL of the inactivated LSDV vaccine preparation intramuscularly (i.m.) at Day 0 and Day 21 post vaccination. Challenge infection was performed at Day 35 post vaccination (Figure 1). Time schedule for Group B was chosen due to the manufacturer’s information that cattle vaccinated with Herbivac LS are complete protected 3 weeks post vaccination [61]. For the inactivated vaccine group, suboptimal vaccine conditions should be simulated to gain better insight in protectivity and efficacy of the inactivated vaccine prototype, which is why challenge infection was performed only 2 weeks after secondary immunization. For challenge infection, the virulent LSDV-“Macedonia2016“ field strain [55,62] with a titer of 10^7^ CCID_50_/mL on MDBK was inoculated s.c. (1 mL) plus intravenously (i.v.) (3 mL) for challenge infection as described before [62].

### 2.3. Vaccine-Efficacy Study

#### 2.3.1. Vaccine Preparation

For development of a safe and efficient inactivated vaccine that can be produced in large scale and to bypass the problems resulting of bovine vaccines produced on bovine cell lines [63], LSDV had to be adapted on a non-bovine cell line. For commercial use, passage history of the used LSDV isolate should also be documented. These criteria were fulfilled by the LSDV-“Serbia” field strain. This strain also provides the advantage of being more closely related to the circulating field strains.

The LSDV-“Serbia“ field strain (internal lab number V/99) was adapted to a non-bovine cell line (Baby Hamster Kidney cells 21 (BHK-21); kindly provided by Zoetis, Olot, Spain) via multiple passaging. For cultivation of cells and propagation of virus, MEM-G (kindly provided by Zoetis, Olot, Spain) with 10% FCS was used. For high virus titers, cells were seeded into 2× T1700 roller bottles and incubated for 2 h at 37 °C and 0.5 rpm for adhesion of cells. Subsequently, 50 mL of frozen-thawed LSDV-“Serbia“ field strain (titer on MDBK cells: 10^5.9^ CCID_50_/mL) were used to infect each roller bottle. After 5 days of incubation at 37 °C/0.5 rpm and further 2 days at 37 °C/1 rpm, the infected cells were harvested using Trypsin-EDTA and centrifuged for 10 min at 5000 rpm. Then, the pellets were re-suspended in TE buffer for concentration of virus and stored at −80 °C until inactivation. Finally, a titer of 10^7.5^ CCID_50_/mL was defined for the pooled virus stock. Subsequently, inactivation with BEI was performed and the inactivated antigen was stored at −80 °C.

Two different adjuvants were used in this study: the same commercially available adjuvant that was used in the first trial (Adjuvant A) and, since Adjuvant A is not authorized in Europe until now, a proprietary adjuvant (Adjuvant B). Due to the respective manufacturers, Adjuvant A belongs to the group of low molecular weight copolymers [64], whereas Adjuvant B consists of a combination of Amphigen, Quil A and Cholesterol (personal communication). In addition, since LSDV-“Serbia” propagated on BHK-21 cell line normally grows with a titer of approx. 10^6^ CCID_50_/mL, two different antigen concentrations were tested for Adjuvant B: 10^7^ CCID_50_/mL before inactivation (as used in the first study) and 10^6^ CCID_50_/mL before inactivation (reflecting propagation of LSDV-“Serbia” on BHK-21 cell line without further processing). A detailed scheme of preparation of the three ready-to-use vaccine preparations is shown in Table 1. Each animal received 2 mL of the respective vaccine preparation for both the primary and the secondary vaccination.

#### 2.3.2. Experimental Design and Challenge Infection

Vaccination was performed i.m. at 0 dpv and 21 dpv as it is standardly performed for inactivated vaccines. Cattle of Group 2A functioned as mock-vaccinated control and received 2 mL 0.063% PBS instead. Cattle of Group 2B were vaccinated with 2 mL of inactivated LSDV (titer approx. 10^7^ CCID_50_/mL)) plus Adjuvant A. Animals of Group 2C and 2D received inactivated LSDV plus Adjuvant B with titers of approx. 10^7^ or 10^6^, respectively. At Day 42 dpv, 2 mL of the virulent LSDV-“Macedonia2016“ field strain with a titer of 10^6.6^ CCID_50_/mL were inoculated i.v. for challenge infection (Figure 2) with regard to recently published results for a robust challenge model of LSDV-“Macedonia2016” field strain [55].

#### 2.3.3. Safety Scoring

Safety of the vaccine prototypes was analyzed 4 h post each vaccination step as well as daily for 14 days post each immunization. Therefore, injection sites were examined carefully regarding adverse effects. In case of visible or palpable reactions in the vaccine-efficacy-study, the exact size (length, width and height) of the reaction was determined. Safety scoring was performed by independent employees; however, study groups were not blinded. In addition, body temperature was measured daily starting from −4 dpv during the vaccine-efficacy study.

## 3. Results

### 3.1. Proof-of-Concept-Study

In general, animals vaccinated with the inactivated prototype vaccine (Group 1C) did not show adverse effects towards immunization, and good clinical protection could be observed after challenge infection, although some animals showed local reactions after inoculation of challenge virus. However, slight viremia and viral shedding could be observed the days after inoculation of challenge material. Cattle vaccinated with Herbivac LS (Group 1B) showed adverse effects after immunization. Here, mainly local reaction at the inoculation side could be observed. Although viremia was detected after immunization, shedding of vaccine virus could not be seen. In contrast, challenge virus DNA was found in different swab samples taken after challenge infection. Animals of the unvaccinated control group (Group 1A) developed severe clinical course typical for LSDV infection after inoculation of virulent challenge virus strain, and strong viremia as well as viral shedding could be observed.

#### 3.1.1. Clinical Reaction

In general, body temperature remained within the normal range after vaccination with inactivated LSDV (Group 1C). After the primary immunization, solely two animals (R-474, 9 dpi; R-476, 17 dpi) showed an increased body temperature (around 39.5 °C) for one day. Following a second immunization, increased body temperatures could be observed for one day in one other cattle (R-552, Day 4) (Appendix A). Additionally, no changes in body temperature could be seen after challenge infection, with two exceptions (R-475, 1 dpc, increased body temperature; R-474, 8 dpc and 9 dpc, fever) (Appendix A and Figure 4). After both the primary and second vaccination, no further adverse effects were detectable in all six cattle (Figure 3C). However, mild (R-475, R-476, R-552) to massive (R-474) reactions at the inoculation site of the challenge virus could be observed starting around 1 dpc and could in some animals still be detected at 28 dpc. Nevertheless, 2 of 6 animals (R-479, R-553) did not show any adverse reaction at all. Moreover, no generalization could be observed after challenge infection in any of the vaccinated cattle (Figure 3C and Figure 4). 

Five to six days after vaccination with the live-attenuated vaccine (Herbivac LS, Group 1B), 3 out of the 6 cattle (R-482, R-543, R-548) developed fever (body temperature >39.5 °C), which lasts for another four to six days. For animal R-482, slightly increased body temperatures could also be observed from Day 17 to Day 19 post vaccination (Appendix A, Figure 4). However, no marked changes in body temperature could be seen after challenge infection. Solely two animals (R-543, R-548) showed a very slight increase in body temperature directly after challenge (Appendix A). Adverse reactions, such as severe local reaction at the inoculation site, could be observed in five of six cattle starting at Day 4 post vaccination and lasting until 17 dpv, leading to clinical reaction scores of 1–5. Only animal R-480 did not develop any side-effects after vaccination. Nevertheless, no generalization could be observed after vaccination with Herbivac LS (Figure 3B and Figure 4). Additionally, no clinical signs could be seen after challenge infection with virulent LSDV in the Herbivac LS vaccination group (Figure 3B and Figure 4).

In the challenge control Group (Group 1A), all cattle showed increased body temperatures after challenge infection, starting around 4 dpc and lasting for approx. 4 to 12 days (Appendix A). Moreover, moderate to severe clinical signs of LSDV with clinical reaction scores between 4 and 10 could be observed, and one animal (R-489) reached the human endpoint and had to be euthanized at 11 dpc (Figure 3A and Figure 4).

#### 3.1.2. Virus Replication and Shedding

An overview of extent and length of viremia and viral shedding is presented in Figure 4. Detailed results of the analyzed viral genome loads at all tested time points and for all taken samples matrices are summarized in Figure 5 and Appendix A.

In the inactivated LSDV-group (Group 1C), viremia could be observed in the EDTA-blood of two cattle after challenge infection (R-474, 9 dpc; R-475, 7 dpc and 9 dpc) (Figure 5C). For these two animals, also the serum samples were tested positive at similar time points and with similar Cq-values (Appendix A). For the remaining cattle of this group, no viremia could be detected in the samples (Figure 5C, Appendix A). Nasal swabs were only positive for viral DNA at 17 dpc and 21 dpc for animals R-476 and R-479 (Figure 5F). Oral swabs were tested positive starting at 14 dpc (R-476, R-479, R-553). At 21 dpc, oral swabs of all six cattle were tested positive for Capripox virus DNA, at 28 dpc only animal R-553 displayed positive results (Appendix A). First positive results in the ocular swabs could be detected at 14 dpc (R-475). At 21 dpc, all cattle scored positive for viral DNA in the ocular swabs. However, at 28 dpc, no Capripox viral DNA could be found in the ocular swabs of all animals (Appendix A).

After vaccination with Herbivac LS (Group 1B), viremia could be observed at 7 and 9 days post immunization in some cattle. EDTA-blood scored positive at 7 dpv in one animal (R-540) and at 9 dpv in three cattle (R-482, R-540, R-548) (Figure 5B). Serum samples of overall four cattle were tested positive for viral genome loads at Day 7 post vaccination (R-482, R-543, R-548) and Day 9 post vaccination (R-540, R-543, R-548) (Appendix A). All oral and nasal swab samples were negative in all animals after vaccination with the live-attenuated vaccine (Figure 5E, Appendix A). In contrast, viremia could not be detected after challenge infection (Figure 5B, Appendix A), but viral shedding could be observed after challenge infection by analyzing nasal (Figure 5E), oral and ocular swab samples (Appendix A). Shedding of viral genome was observed to be stronger in the oral and ocular swabs compared to the nasal swabs (Figure 5E, Appendix A).

Cattle of the challenge control group (Group 1A) showed a stronger viremia as well as higher viral genome loads in the swab samples compared to both vaccinated groups (Figure 4). At 7 dpc, viremia could already be observed in the EDTA-blood of all six animals and with lower Cq-values compared to the other groups (Figure 5A). In addition, viral genome could be observed in serum samples of all cattle, except animal R-556 (Appendix A). The first positive nasal swab was detected at 2 dpc (R-489). At 7 dpc, 9 dpc as well as 12 dpc, approx. half of the cattle tested positive in the nasal swabs. At 14 dpc and 21 dpc, nasal swabs of all remaining five cattle were positive for viral genome in the PCR-analysis, and nasal swabs of 4 of 5 cattle remained positive for Capripox virus genome until the end of the study (Figure 5D). Similar patterns could be observed for oral swabs with the first positive oral swabs at 9 dpc (R-487, R-489) (Appendix A). The first positive ocular swabs were observed later compared to nasal and oral swabs, at 12 dpc (R-477, R-487). Two days later, at 14 dpc, ocular swabs of all remaining five cattle were tested positive for viral genome. However, at 28 dpc, no viral genome could be detected in the ocular swabs of all cattle of the challenge control group (Appendix A).

#### 3.1.3. Serological Response

Cattle vaccinated with inactivated LSDV (Group 1C) were immunized at Day 0 of the animal trial and Day 21 post primary vaccination (in the following defined as dpv). Challenge infection was performed at Day 35 of the animal trial (14 days after secondary vaccination). Using the SNT, first positive results in the serological analyses could already be observed at 21 dpv in all animals except R-475 (Figure 6C), whereas the DA ELISA scored negative in all six cattle at 21 dpv (Figure 6F). However, all cattle showed positive results in the DA ELISA as well as in the SNT at the day of challenge infection (Figure 4 and Figure 6C,F).

Challenge infection of the cattle vaccinated with the live-attenuated vaccine (Group 1B) was performed at 21 dpv. At this time point, 5 of 6 animals were positive in the SNT, solely for animal R-480 no neutralizing antibodies could be observed. However, at 7 dpc, positive results in the SNT could be obtained also for R-480 (Figure 6B). A similar pattern can be seen for the results of the DA ELISA. At 0 dpc, the DA ELISA was positive for all animals except R-490, which in turn also scored positive at 14 dpc for the first time (Figure 4 and Figure 6E).

In the challenge control group (Group 1A), first serological response was observed at 14 dpc using the SNT method. Nevertheless, at this time point, all remaining five animals were tested positive for neutralizing antibodies (Figure 6A). Additionally, the DA ELISA scored positive for two (R-484, R-556) of the five cattle at 14 dpc, whereas the other three cattle showed positive results in the DA ELISA from 21 dpc on (Figure 4 and Figure 6D).

### 3.2. Vaccine-Efficacy Study

Animals immunized with the Adjuvant A-containing vaccine (Group 2B) did not show adverse effects after primary and secondary immunization and were completely protected against challenge infection and neither viremia nor viral shedding could be detected. For animals of both Adjuvant B-containing vaccine groups (Group C and Group D), local reaction at the inoculation side of the vaccine could be seen. In addition, the cattle were only partially protected against severe challenge infection, and some animals developed skin lesions from which replicable challenge virus could be isolated. However, none of the cattle of these groups showed generalized disease, and neither viremia nor viral shedding could be observed in 11 out of 12 animals. PBS-inoculated cattle of the challenge control group (Group 2A) developed severe lumpy skin disease, and observed pattern of viremia as well as viral shedding was typical for experimental infections with LSDV-“Macedonia2016” field strain.

#### 3.2.1. Clinical Reaction and Safety Observation

In the Adjuvant A 10^7^-group (Group 2B), no adverse effects could be detected for 14 days after the primary immunization in all six animals nor in five cattle for 14 days after secondary vaccination. Only animal R/848 showed very mild local reaction at the inoculation site at Day 2 and Day 3 after the second immunization (Figure 7). Body temperatures remained within the normal range after both the primary and the secondary vaccination. Additionally, no significant changes in body temperatures were observed after challenge infection. Solely animal R/770 showed a fever peak at 1 dpc (40.0 °C), but normalized quickly (Figure 7, Appendix A). Interestingly, no clinical reactions could be observed for 28 days after challenge infection with virulent LSDV in the whole group (Figure 7 and Figure 8B).

Contrarily, adverse effects could be detected after primary and secondary immunization in all animals of both Adjuvant B-groups. After primary vaccination with Adjuvant B 10^7^ (Group 2C), four of six cattle showed local reactions at the inoculation site for 6 to 11 days. After the second vaccination, four cattle showed only slight local reactions at the inoculation site lasting for one or two days, whereas two animals again had moderate reactions at the site of inoculation for 10 days (Figure 7). In the Adjuvant B 10^6^ group (Group 2D), adverse reactions after the primary immunization were reduced compared to Adjuvant B 10^7^, in detail, local reactions at the inoculation site could only be detected in two cattle for a single day. However, after secondary vaccination, mild to moderate reactions at the inoculation site could be observed in all six animals, lasting for 2 to 11 days (Figure 7). For nearly all animals that were vaccinated with the Adjuvant B-containing vaccine, independent of the antigen concentration, a fever peak could be observed one day after primary and secondary vaccination as well as one day after challenge infection. Besides this, no marked decrease or increase of body temperatures could be observed in any of the Adjuvant B-groups during the whole study (Figure 7, Appendix A+D). After challenge infection, mild clinical reactions could be seen in three (R806, R/464, R/457) out of six cattle of the Adjuvant B 10^7^ group (Figure 7 and Figure 8C), namely sporadic swelling of lymph nodes and formation of sporadic or multiple skin lesions. Interestingly, these lesions were different from the typical LSDV-lesions (Figure 9), but scored positive in the pan-Capripox real-time qPCR (Table 2), and virus isolation on MDBK cells was successful for these samples (Figure 10) (detailed description: Analyses of skin nodes). Overall, the clinical reaction scores were not higher than 3 (R/464, 12–14 dpc) (Figure 8C). In the Adjuvant B 10^6^ group, five cattle remained inconspicuous after challenge infection, solely animal R/782 showed slightly reduced activity from 7 dpc until 9 dpc (Figure 8D). In contrast, animal R/833 displayed swelling of multiple lymph nodes and developed numerous lesions similar to those seen in the Adjuvant B 10^7^ group (Figure 9), leading to a clinical reaction score of up to 4 (8 dpc–9 dpc and 11 dpc–18 dpc) (Figure 7 and Figure 8D).

In the control group (Group 2A), inoculation of PBS for control of mechanical influence on adverse effects during the vaccination process did not lead to any detectable reaction at the inoculation sites of all six cattle (Figure 7). Body temperature of the unvaccinated control group did not show any marked changes after both the first and second inoculation of PBS. However, as expected, increased body temperatures and fever occurred in all six cattle starting around 4 dpc, with four cattle reaching >40.0 °C (R/792, R/783, R/776, R/800) (Figure 7, Appendix A). Moreover, first clinical signs typical for LSDV infections could be observed as early as 4 dpc. Although one animal only showed a mild clinical course (R/817, maximum clinical reaction score 3), and one cattle developed moderate LSD (R/779, maximum clinical reaction score 6), four animals got severely affected by LSD (clinical reaction score of 9 and 10, respectively) and had to be euthanized at 9 dpc due to ethical reasons (Figure 7 and Figure 8A).

#### 3.2.2. Analyses of Skin Nodes

After challenge infection, some of the cattle previously immunized with the Adjuvant B-containing vaccine (groups 2C and 2D) developed skin nodes. These were different from typical LSDV-lesions in shape and appearance (Figure 9). These skin nodules appeared to be subcutaneously and seemed to be clearly separated. In the challenge control group (Group 2A), typical LSDV-lesions appeared either sporadically on the neck and sometimes also in the udder region or generalized all over the body. In contrast, those different nodules observed in the Adjuvant B-groups appeared in lower number and mainly on the neck and shoulder of the affected animals.

To verify that these nodules were caused by LSD infection, samples were taken, and viral genome load was determined. In addition, skin samples of affected cattle of the challenge control group were also tested to function as positive control for sampling, processing of samples, and analyses. A skin sample of one animal of the Adjuvant A 10^7^ group (Group 2B) was taken as negative control. Using the pan-Capripox real-time qPCR, viral genome could be detected in almost all skin node samples of all animals either of the control group or of the Adjuvant B-containing vaccine groups. Solely the cattle vaccinated with the Adjuvant A-consisting vaccine displayed negative results in the qPCR (Table 2).

Moreover, virus isolation on MDBK from the samples that were positive for Capripox virus genome was successful (Figure 10), giving evidence for those nodes contain replicable and infectious virus particles.

#### 3.2.3. Virus Replication and Shedding

For Adjuvant A 10^7^-vaccinated cattle (Group 2B), neither cell-associated nor cell-free viremia could be detected in all EDTA-blood and serum samples at all time points. Additionally, all nasal swabs of the group were negative for viral genome (Figure 7 and Figure 11B,F, Appendix A). Interestingly, this applies also for the Adjuvant B 10^6^ group (Group 2D) (Figure 7 and Figure 11D,H, Appendix A), but not for the Adjuvant B 10^7^-vaccinated group (Group 2C). In the latter, for one cattle (R/464) viremia could be detected at 7 dpc (EDTA-blood, Cq 34.3) and 10 dpc (EDTA-blood, Cq 24.4; serum, Cq 37.0) (Figure 7 and Figure 11C,G, Appendix A).

In the mock-vaccinated control animals (Group 2A), first viremia could be observed at 5 dpc in the EDTA-blood of animals R/792 (Cq 37.1) and R/800 (Cq 32.2) (Figure 7 and Figure 11A). In the following days, viremia could be observed in all cattle of the challenge control group (Figure 7 and Figure 11A, Appendix A). Nasal swabs started to be positive at 7 dpc in three out of six animals with Cq-values around 35 (Figure 11E). Those four cattle that had to be removed from the trial due to ethical reasons at 9 dpc, displayed positive results in the pan-Capripox real-time qPCR in all three tested matrices at the day of euthanasia. Both remaining animals showed only mild viremia and no (R/779) or slight (R/817) viral shedding (Figure 7 and Figure 11A,E, Appendix A).

#### 3.2.4. Viral Genome Load in Certain Organ Samples

Cervical lymph nodes of all vaccinated cattle, independently of the vaccine type, were negative for Capripox virus genome. In contrast, cervical lymph nodes of five animals of the control group (Group 2A) scored positive in the pan-Capripox real-time qPCR with Cq-values of four animals relatively low (between 21.9 and 28.4). Although in the euthanized animals the mediastinal lymph nodes were also positive for viral genome (Cq-values from 23.3 to 35.2), no Capripox virus genome could be found in the mediastinal lymph nodes of both surviving animals (R/817, R/779). However, the mesenterial lymph nodes of all unvaccinated cattle were tested negative for viral genome (Table 3).

#### 3.2.5. Serological Response

At 21 dpv (time of secondary immunization), serum samples of two (R/788, R/784) of the six Adjuvant A 10^7^-vaccinated cattle (Group 2B) gave positive results in the DA ELISA, and at the day of challenge infection (42 dpv), the performed DA ELISA was positive for all six animals of this Group. After challenge infection, antibody levels of two animals (R/770, R/799) increased according to the ELISA results, whereas the S/P% ratio remained in a constant range for the other four cattle of Group 2B (Figure 7 and Figure 12B). Only one Adjuvant B-vaccinated animal (R/806, Adjuvant B 10^7^) had detectable amounts of antibodies at 21 dpv by DA ELISA. Nevertheless, at 42 dpv, serum samples of all animals of both Adjuvant B-groups (groups 2C and 2D) gave positive results in the DA ELISA. After challenge infection, S/P% ratio of nearly all animals increased over time (Figure 7 and Figure 12C,D). In the unvaccinated control group (Group 2A), only the two surviving animals developed detectable amounts of antibodies, starting around 17 dpc (R/817) and 21 dpc (R/779) (Figure 7 and Figure 12A). Missing antibody detection for the four euthanized cattle is due to the early time point of euthanasia (9 dpc).

Additionally, ELISA titers as well as SNT titers of all serum samples of 42 dpv and 28 dpc were analyzed in comparison. Differences in the titers between 42 dpv and 28 dpc were defined as ≥ 2 log_2_. Otherwise (difference ≤ 1 log_2_), titers were defined as equal. From 42 dpv until 28 dpc, ELISA titer increased in only two (R/770, R/839) out of six cattle of the Adjuvant A 10^7^ group (Group 2B), whereas it stayed equal for the other four cattle. Furthermore, SNT titers remained stable along all this time in all animals immunized with the Adjuvant A-containing vaccine (Table 4, Figure 7). In the Adjuvant B 10^7^ group (Group 2C), ELISA titer increased in 4/6 animals from 42 dpv until 28 dpc and remained equal only in the other two cattle (R/433, R/547) of this group. However, SNT titers were stable for five cattle, solely animal R/464 showed an increase in the SNT titer from 42 dpv until 28 dpc (Table 4, Figure 7). A similar pattern could be observed for the animals vaccinated with the Adjuvant B 10^6^-containing vaccine (Group 2D). Here, the ELISA titer of only animal R/815 remained equal during this time period, for all other animals of this group a clear increase in ELISA titer could be observed. Nevertheless, in the SNT, a difference in the titer between 42 dpv and 28 dpc could only be detected for R/833. For the other animals of the Adjuvant B 10^6^ group, an equal SNT titer could be observed at 42 dpv and 28 dpc (Table 4, Figure 7).

## 4. Discussion

Due to their properties, inactivated vaccines are a great and safe tool for the control of different diseases. Contrarily to most modified live vaccines, they are non-replicating, and therefore do not compromise the disease-free status of the respective countries [14,20]. Although they have been described as insufficiently immunogenic for protection against Capripox viruses [47], a few studies revealed promising results for inactivated vaccines against the three Capripox virus species [45,46,48,49,50,51,52].

In our first animal trial of vaccination against LSDV, an Adjuvant A-adjuvanted inactivated prototype vaccine, based on the LSDV-“Neethling vaccine“ strain, was compared with the live-attenuated Herbivac LS vaccine regarding adverse effects and protection of the cattle against i.v. and s.c. challenge infection with the virulent LSDV-“Macedonia2016“ field strain. This study design is very similar to the concept of the recently published study of Hamdi et al. (2020). They also compared a BEI-inactivated attenuated LSDV-“Neethling” strain with a live vaccine against LSDV [46]. In our proof-of-concept study, no side-effects were detectable in the vaccinated cattle after both the primary and secondary immunization with the inactivated LSDV (Figure 3, Appendix A). In contrast, Hamdi et al. observed slight increase in body temperatures and local reactions at the inoculation site a few days post vaccination with their inactivated prototype vaccine, which persisted for several days [46].

Since inactivated pathogens are unable to replicate, no viremia or viral shedding could be observed in the cattle vaccinated with inactivated LSDV in the weeks post immunization during our study (Figure 5, Appendix A). Additionally, all animals had seroconverted before the day of challenge infection, which was verified by ELISA as well as SNT (Figure 4 and Figure 6). Contrary to the results of Hamdi et al., who achieved complete clinical protection after vaccination with inactivated attenuated LSDV-“Neethling” strain and neither viremia nor oral shedding of challenge virus could be observed [46], cattle of our study immunized with inactivated LSDV were protected only partially against challenge infection. Two animals did not show any reaction at all, and only mild or massive local reactions at the site of subcutaneous inoculation of challenge virus could be observed for three and one cattle, respectively. However, no generalized forms of LSDV could be seen after challenge infection (Figure 3 and Figure 4). Viremia could only be detected in 2 out of 6 animals after challenge infection. Although all six cattle shed viral DNA via oral and ocular fluid, nasal swab samples were only positive for 2 of 6 cattle (Figure 4 and Figure 5, Appendix A). Furthermore, an increase of the antibody reactivities could be detected in the days following the inoculation of challenge virus (Figure 6).

Taken together, these findings provide evidence that inactivated LSDV can protect cattle from severe clinical course of LSD. This is also consistent with a recently published study of Boumart et al., in which the efficacy of an inactivated vaccine against SPPV was evaluated. Here, local reactions at the inoculation site of virulent SPPV challenge virus but no clinical signs typical for SPPV infections could be observed in the respective sheep [45]. In contrast to the inactivated LSDV-group, some of the Herbivac LS-vaccinated cattle showed fever for several days as well as severe local reaction at the inoculation site for days to weeks after vaccination. However, no generalization could be observed due to the live-attenuated vaccine (Figure 3 and Figure 4, Appendix A). These findings are consistent with previously published observations describing increased body temperatures [33,36,38] and local reactions after inoculation [23,38,39,40], but no generalization or mortalities [33]. Due to the characteristics of live-attenuated vaccines, viremia could be detected in cattle vaccinated with Herbivac LS using molecular analysis (Figure 5, Appendix A). This is not unexpected, since the vaccine virus replicates in the host and detection of vaccine virus DNA in whole blood samples or buffy coat of vaccinated cattle was described previously [33,39]. Nevertheless, neither vaccine virus nor vaccine virus DNA was shed via nasal, oral and ocular fluids during our study (Figure 5, Appendix A). This is similar to the results obtained from Katsoulos et al., who also did not find vaccine virus in the saliva after vaccination with a LSDV-“Neethling vaccine“ derivative [39]. In this Herbivac LS-vaccinated group, all animals also seroconverted before challenge infection, although one animal was negative in the ELISA and one other cattle displayed negative results in the SNT at this time point (Figure 4 and Figure 6). For all animals of this group, complete clinical protection could be observed after challenge infection (Figure 3 and Figure 4, Appendix A), which is consistent with previous reports that vaccination with homologous live-attenuated vaccines provides good protection against challenge infection [28,30,36]. Moreover, viremia could not be detected in EDTA-blood or serum samples at all different time points of sampling after challenge infection during our study (Figure 5, Appendix A). Nevertheless, viral DNA was found in nasal swabs as well as ocular swabs in five out of six cattle and in oral swabs of all animals at different days post challenge infection, indicating replication and shedding of challenge virus (Figure 4 and Figure 5, Appendix A). Compared to the inactivated LSDV-group, shedding of Capripox virus DNA seemed to be slightly increased in the Herbivac LS-group (Figure 5, Appendix A). In addition, an increase of antibody titers could be observed from the day of challenge until 28 dpc, which is similar to the findings of the inactivated LSDV-group (Figure 6). The used challenge model turned out to be highly effective. All cattle of the challenge control group developed clinical signs typical for LSDV infections, but with different severity. In this group, moderate to severe clinical course could be observed, and one animal had to be euthanized before the end of the study due to ethical reasons (Figure 3 and Figure 4, Appendix A). These findings correlate well with the results of one of our previous studies in which we analyzed the challenge virus strain LSDV-“Macedonia2016“ in cattle. Here, also different severities of LSD were observed in the inoculated cattle and three of six animals had to be euthanized [62].

Due to these consistent outcomes, experimental infection with LSDV-“Macedonia2016“ can be used as a robust, strong, and effective challenge model for vaccine testing. Nevertheless, subcutaneous inoculation led to local reactions at the inoculation sites in almost all animals. In summary, the results of the proof-of-concept study revealed no side-effects after vaccination, good clinical protection after a strong challenge infection and similar to slightly reduced shedding of challenge virus in the inactivated LSDV-vaccinated cattle compared to the Herbivac LS-cattle. This confirms the results recently described [46] that control of LSDV using an inactivated vaccine, similar to SPPV [45,48,49] and GTPV [50,51,52], might be possible.

A significant advantage of both of our prototype vaccines is the adaptation of vaccine virus on permanent production cell lines compared to propagation of vaccine virus on primary cells as described for the inactivated vaccine of Hamdi et al. [46]. Since propagation of vaccine virus in permanent cell lines can be performed more standardized, these cells are preferred for commercial vaccine production. Due to negative experiences with an inactivated BVD-vaccine propagated on a bovine cell line, resulting in Bovine Neonatal Pancytopenia in calves after intake of colostrum of BVD-vaccinated dams [63], propagation of LSDV for development of an inactivated vaccine should be performed on non-bovine cell lines. We therefore chose an adherent production cell line provided by Zoetis for the following experiments. Additionally, protection efficacy of another LSDV isolate, a Serbian field strain of LSDV, was examined in the second study. Moreover, two different adjuvants (the same low molecular weight copolymer adjuvant as used in the first trial (Adjuvant A), and a proprietary adjuvant (Adjuvant B)) were compared as well as two different antigen concentrations (infectious virus titer before inactivation 10^7^ and 10^6^ CCID_50_/mL, respectively) using Adjuvant B (Table 1, Figure 2). As shown before [55], for experimental infection of cattle with LSDV intravenous inoculation is an effective route to cause clinical and in many cases generalized LSD. We determined the minimum infective dose of LSDV-“Macedonia2016“ when inoculated only intravenously in a recent study [55]. Considering this, 2 mL of LSDV-“Macedonia2016“ with a titer of 10^6.6^ CCID_50_/mL were inoculated intravenously for challenge infection.

Interestingly, vaccination with the Adjuvant A-containing vaccine prototype did not reveal any adverse effects nor increased body temperature for 14 days following primary and secondary immunization, respectively, whereas cattle vaccinated with both Adjuvant B-containing vaccines displayed a one-day fever peak as well as mild to massive local reactions at the inoculation site for several days (Figure 7 and Figure 8, Appendix A). This is similar to the pattern observed after inoculation of infectious challenge virus. Here, increased body temperature could only be observed in one animal of the Adjuvant A 10^7^ group and in almost all Adjuvant B-vaccinated animals independently of the antigen concentration (Figure 7, Appendix A). This is not unexpected, since slight increase of body temperature and local reaction a few days post vaccination with an oily adjuvanted inactivated attenuated LSDV-“Neethling” strain have been described previously [46].

Taken together, these findings suggest that the choice of the adjuvants is of significance for prevention or at least reduction of side-effects after vaccination. However, selecting the right adjuvant is also important for the protection effect of inactivated vaccines against LSDV. In the Adjuvant A 10^7^ group, neither clinical reactions (Figure 7 and Figure 8, Appendix A) nor viremia and viral shedding (Figure 7 and Figure 11, Appendix A) could be observed during the study observation period, indicating complete protection against LSDV infection. This is in accordance with results previously described, in which an oily adjuvanted prototype vaccine was able to protect cattle completely from challenge infection with virulent LSDV [46]. Conversely, Adjuvant B-containing vaccine prototypes protected the cattle against severe clinical course of LSD as well as generalized forms of the disease, but symptoms typical for LSDV infections could be observed in some of the vaccinated animals after challenge infection (Figure 7 and Figure 8). Additionally, replicable and infectious virus was detected in the occurring skin nodules (Table 2, Figure 9 and Figure 10). However, viremia and nasal shedding of challenge virus could be prevented in almost all animals of both Adjuvant B-groups (only exception was viremia of R/464) (Figure 7 and Figure 11, Appendix A), indicating at least good partial protection against LSDV. Nevertheless, the presence of challenge virus in the skin nodes of Adjuvant B-vaccinated cattle could possibly be a source for mechanical transmission of challenge virus from vaccinated and afterwards infected animals to naïve cattle by biting insects. Surprisingly, antibody titers in the SNT remained equal in all Adjuvant A-animals as well as almost all Adjuvant B-animals, when comparing 0 dpc and 28 dpc. A similar result could be observed for the ELISA titer of the Adjuvant A-group, in which the ELISA antibody titers in 4 out of 6 animals were equal at 0 dpc and 28 dpc. In contrast, ELISA titers increased in a total of nine of twelve cattle of the Adjuvant B-groups (Table 4). However, no difference between both Adjuvant B-groups could be detected regarding side-effects after vaccination, clinical protection against challenge infection, viremia, shedding of virus, and seroconversion.

As expected, according to results of previous studies with LSDV-“Macedonia2016“, the challenge model again worked very robustly in the vaccine-efficacy study. Intravenous inoculation with challenge virus caused LSD in all six cattle of the challenge control group. Four animals developed generalized LSD and had to be euthanized due to ethical reasons before the end of the trial, whereas only two cattle displayed a moderate clinical course and recovered from the infection (Figure 7 and Figure 8, Appendix A). Taken together, these data support and enhance the finding of the proof-of-concept study as well as of the results of Hamdi et al. (2020) that an inactivated LSDV vaccine can protect cattle from LSD. Furthermore, protection efficacy does not seem to depend mainly on the virus strain used for an inactivated vaccine, as both the LSDV-“Neethling vaccine“ strain and the LSDV-“Serbia” field strain were able to protect cattle from challenge infection. Nevertheless, cattle of the proof-of-concept study (inactivated LSDV-“Neethling vaccine” strain) showed at least partial clinical protection against strong challenge, whereas all cattle of the Adjuvant A-group (Group 2A, inactivated LSDV-“Serbia“ field strain) were completely protected against LSDV challenge infection. The route of inoculation and the slightly reduced dose of the challenge virus might explain differences in the efficacy of protection. Although challenge infection was performed intravenously plus subcutaneously in the proof-of-concept study, cattle of the second study were inoculated intravenously only. Effect of subcutaneous inoculation of virus on local reaction at the inoculation site seems highly possible under these circumstances. In addition, challenge infection was set already at 14 days after secondary immunization in the proof-of-concept study, whereas in the vaccine-efficacy study, challenge infection was performed as late as 21 days after the second immunization. These additional seven days between the second vaccination and challenge infection also might influence the strength of protection against the virulent challenge virus. Nevertheless, the inactivated LSDV-“Serbia” vaccine prototype also prevented viremia as well as viral shedding, and thereby fits nicely to the previous report [46], but is in contrast to the results obtained in the first study in which shedding of viral DNA could be found in nasal, oral, and ocular swabs after challenge infection. Taking into consideration that an oily adjuvanted inactivated attenuated LSDV-“Neethling” strain was able to induce complete protection against challenge infection [46], impact of the used LSDV strain for preparation of the respective prototype vaccine seems unlikely. Since also antibody reactions remained equal between 0 dpc and 28 dpc in Adjuvant A 10^7^-vaccinated cattle of the vaccine-efficacy study, development of sterile immunity in cattle vaccinated with inactivated LSDV-“Serbia” plus Adjuvant A can be assumed.

Our results in combination with the study recently published [46] indicate that choice of the used adjuvant is more important for protection efficacy of an inactivated LSDV vaccine than the used virus strain or the antigen concentration. This might be explained by the characteristics of the different adjuvants and the immune response they induce in the host. It is long known that Capripox viruses stimulate both humoral and cellular immune response [34,35,65]. However, no correlation has been detected between the levels of neutralizing antibodies and the immune status of the animals [23,47]. Additionally, it is known that Capripox virus infections or vaccination with live-attenuated vaccines effectively stimulate cell-mediated immune response of the animals [34,37,38,66]. According to the manufacturer, Adjuvant A is a low molecular weight copolymer that is able to efficiently stimulate humoral as well as interferon gamma responses, which makes it a great choice for vaccines where cellular immune response is highly required [64]. These characteristics together with our results might explain the better efficacy of the Adjuvant A-based vaccine prototype compared to the Adjuvant B-based vaccine prototypes. Since the choice of the adjuvant turned out to be more important in this trial than the antigen concentration, further studies must be performed focusing on the minimum protective antigen amount that is necessary to achieve complete protection against LSDV. In addition, future vaccine studies should include examination of cellular immune response towards immunization with inactivated or live-attenuated vaccines against LSDV. Furthermore, duration of immunity after immunization with inactivated vaccines against LSDV must be examined in the future.

Unfortunately, serological DIVA (Differentiating Infected from Vaccinated Animal) assays to differentiate between vaccinated and infected animals are missing. However, this is also the case for animals vaccinated against LSDV with live-attenuated vaccines due to the fact that all capripox viruses are antigenically indistinguishable [30], displaying a general problem of vaccination against Capripox viruses. Although inactivated vaccines are more expensive than live-attenuated vaccines due to higher cost requirement and necessity of at least two immunizations as well as possible re-vaccination after certain periods of time, they provide a great advantage for preventive usage in disease-free countries.

## 5. Conclusions

In conclusion, our results using an inactivated vaccine prototype against LSDV are very promising and in accordance with findings previously described dealing with inactivated Capripox virus vaccines [45,46,48,49,50,51,52]. Complete clinical protection could be achieved with a BEI-inactivated and copolymer-adjuvanted LSDV-“Serbia” vaccine without causing adverse effects after vaccination. Moreover, our data indicate that this prototype vaccine was able to generate sterile immunity in the vaccinated cattle.

## Figures and Tables

**Figure 1 vaccines-09-00004-f001:**
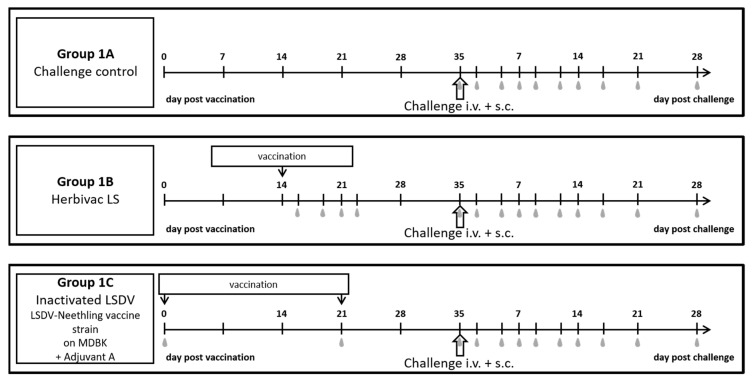
Experimental design of the performed proof-of-concept study. Cattle of Group 1A were not vaccinated. Cattle of Group 1B received the commercially available Herbivac LS live-attenuated vaccine at Day 14 of the animal trial. Cattle of Group 1C were vaccinated twice (Day 0 and Day 21 of the animal trial) with inactivated LSDV-“Neethling vaccine“ strain (titer before inactivation 10^7.4^ CCID_50_/mL) propagated on MDBK cells and using Adjuvant A. Challenge virus was inoculated i.v. and s.c. at Day 35 of the animal trial. Grey marks display sampling days.

**Figure 2 vaccines-09-00004-f002:**
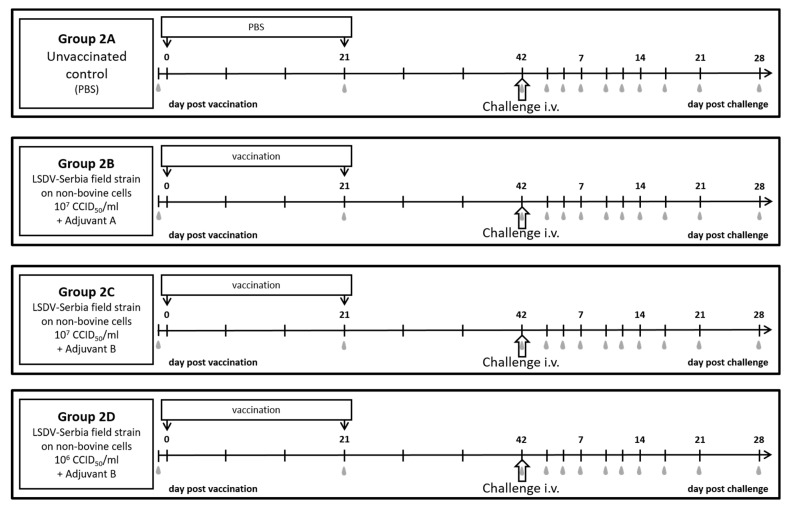
Experimental design of the performed vaccine-efficacy study. Cattle of Group 2A were not vaccinated but received PBS instead. The cattle of the other groups were vaccinated with inactivated LSDV-“Serbia“ field strain propagated on a non-bovine cell line. Group 2B received a vaccine with a virus infectious titer before inactivation of 10^7^ CCID_50_/mL and Adjuvant A. For Group 2C and 2D, the inactivated virus was used with different titers (10^7^ CCID_50_/mL and 10^6^ CCID_50_/mL, respectively) and Adjuvant B. Grey marks display sampling days.

**Figure 3 vaccines-09-00004-f003:**
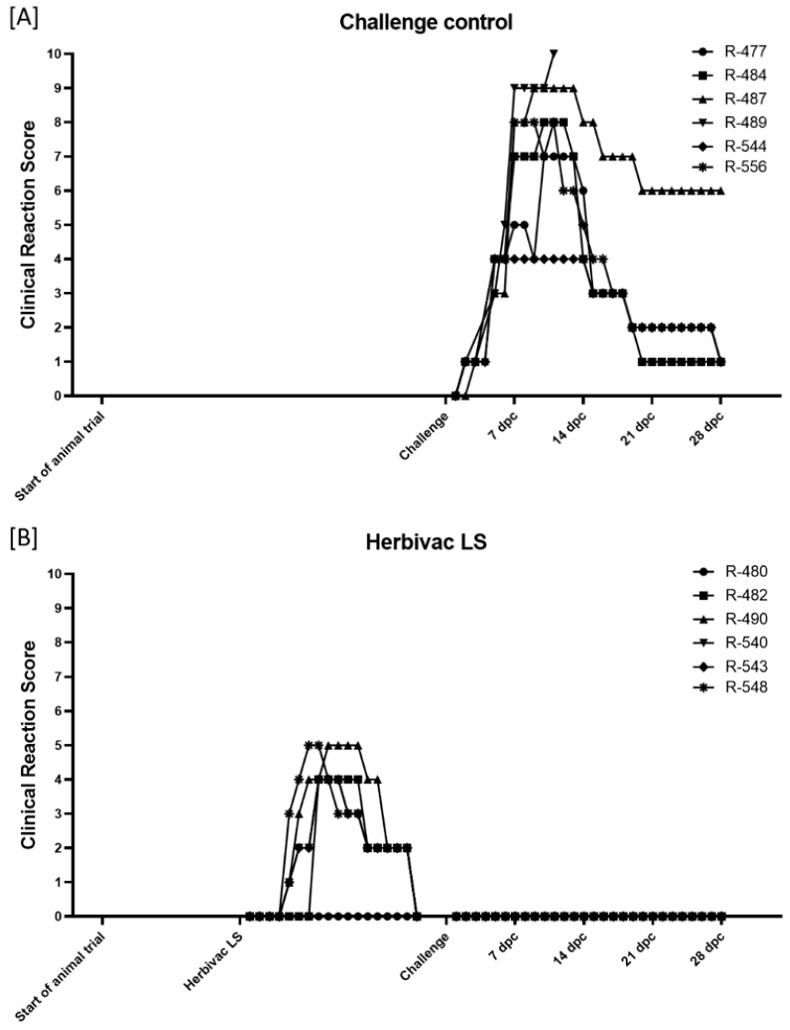
Clinical reaction score of the cattle during the proof-of-concept animal trial. Clinical reaction score was measured daily starting a few days before first vaccination with inactivated LSDV until 28 dpc. (**A**) Cattle of Group 1A were left unvaccinated and served as challenge control group. (**B**) Cattle of Group 1B received the commercially available life-attenuated vaccine “Herbivac LS”, and (**C**) cattle of Group 1C were vaccinated with an inactivated vaccine prototype on basis of LSDV-“Neethling vaccine” strain.

**Figure 4 vaccines-09-00004-f004:**
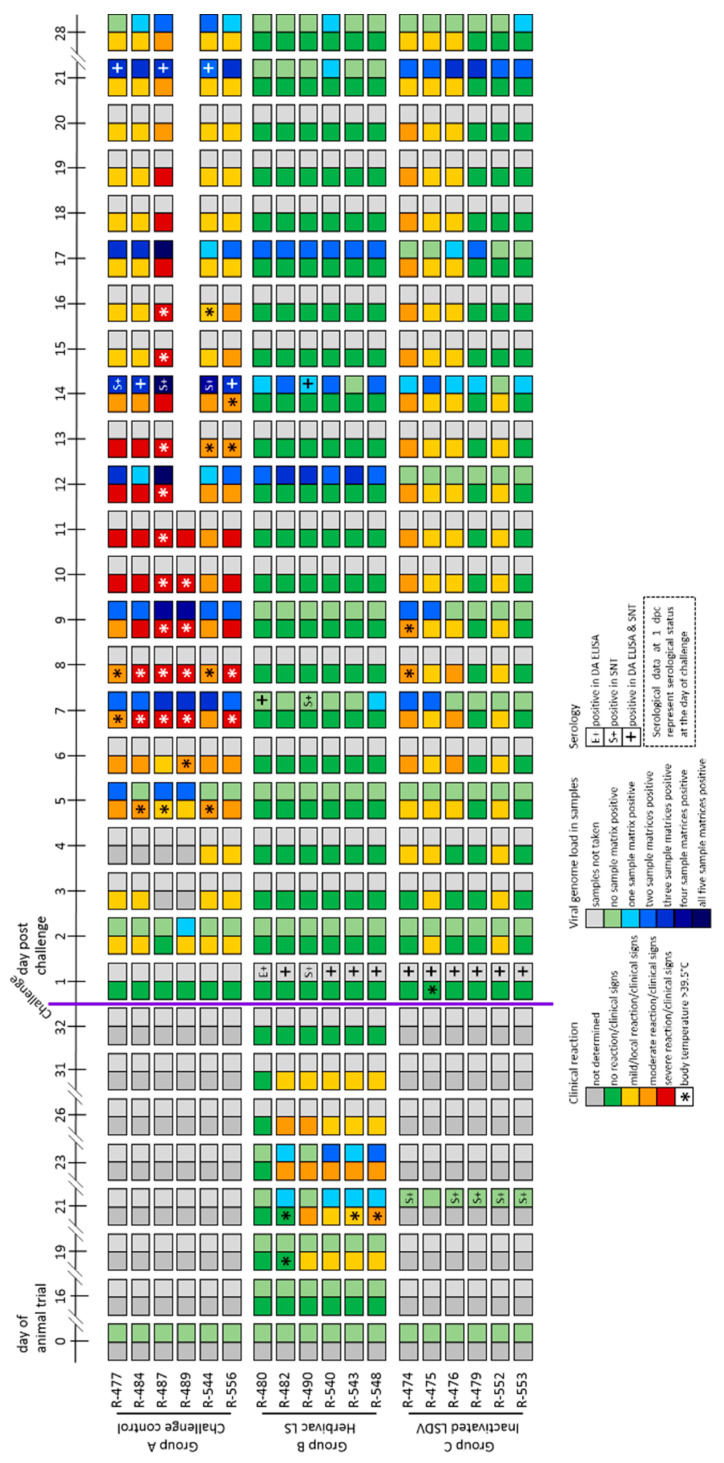
Summarized presentation of adverse reactions after immunization, clinical reaction, viral genome load in different samples matrices and serological examination of the proof-of-concept-study. Adverse reactions are shown at representative days post vaccination. Furthermore, strength of clinical reaction and rise in body temperature after immunization or challenge infection are presented (left squares of each day) for each animal. Additionally, extend of viremia and shedding of viral genome as well as serological data (right squares of each day) following challenge infection are shown for each individual.

**Figure 5 vaccines-09-00004-f005:**
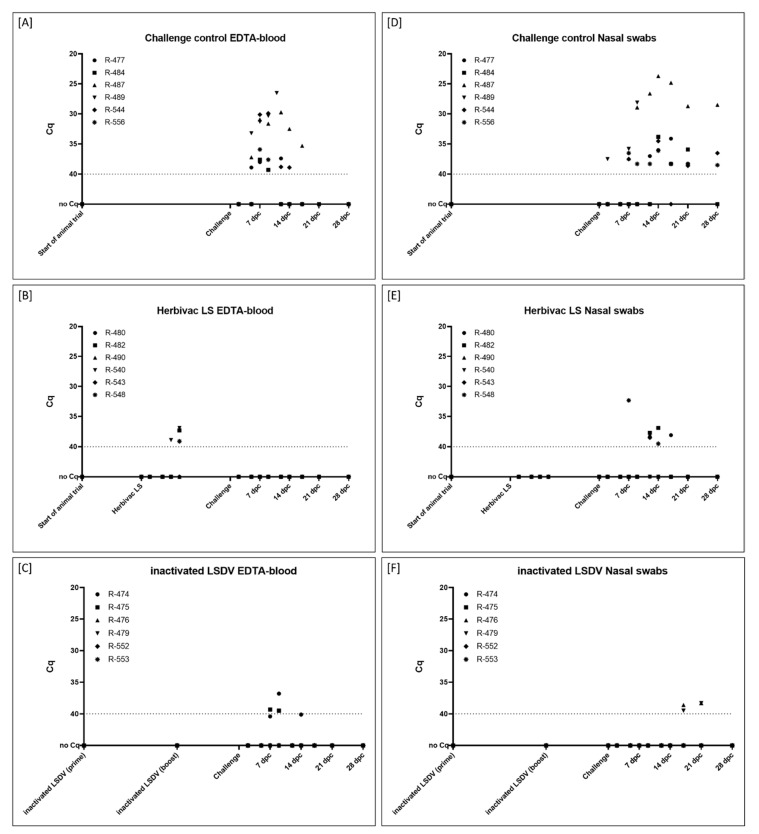
Viral genome load in (**A**–**C**) EDTA-blood and (**D**–**F**) nasal swab samples taken during the proof-of-concept study. (**A**+**D**) Cattle of Group 1A were left unvaccinated and served as challenge control group. (**B** + **E**) Cattle of Group 1B received the commercially available life-attenuated vaccine “Herbivac LS”, and (**C** + F) cattle of Group 1C were vaccinated with an inactivated vaccine prototype on basis of LSDV-“Neethling vaccine” strain. Samples were taken at defined time points during the animal trial and analyzed regarding their viral genome load. Cut-off was defined at Cq 40.0.

**Figure 6 vaccines-09-00004-f006:**
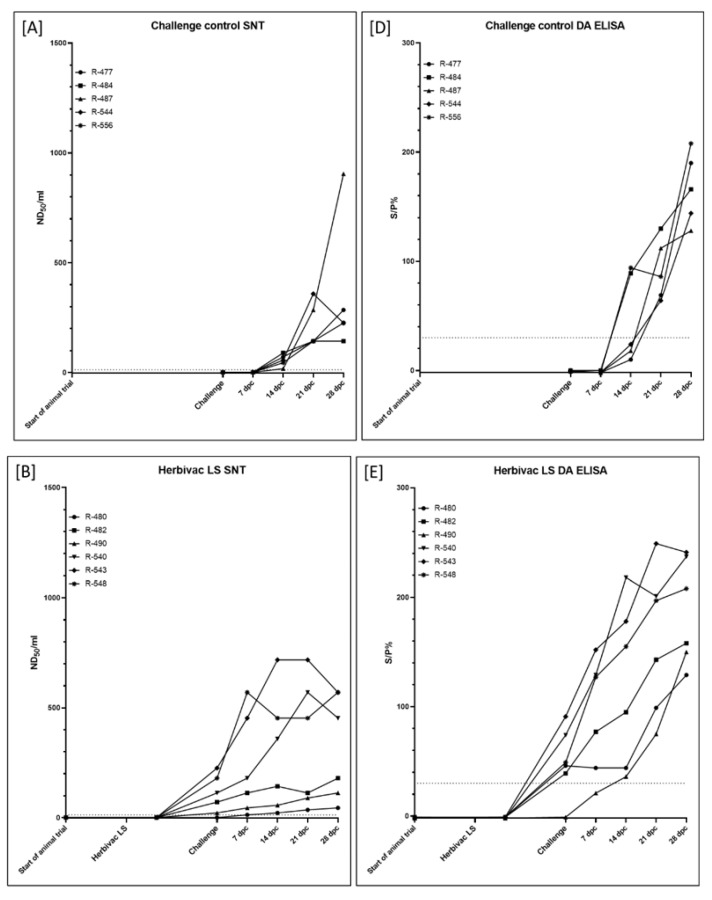
Seroconversion of the cattle of the proof-of-concept animal trial. Serum samples taken at certain time points during the study were analyzed using (**A**–**C**) the serum neutralization test (SNT) and (**D**–**F**) the DA ELISA. (**A** + **D**) Cattle of Group 1A were left unvaccinated and served as challenge control group. (**B** + **E**) Cattle of Group 1B received the commercially available life-attenuated vaccine “Herbivac LS”, and (**C** + **F**) cattle of Group 1C were vaccinated with an inactivated vaccine prototype on basis of LSDV-“Neethling vaccine” strain. Samples in the SNT were defined positive at ND_50_/mL ≥ 13, whereas in the ELISA samples with S/P% ≥ 30 are positive.

**Figure 7 vaccines-09-00004-f007:**
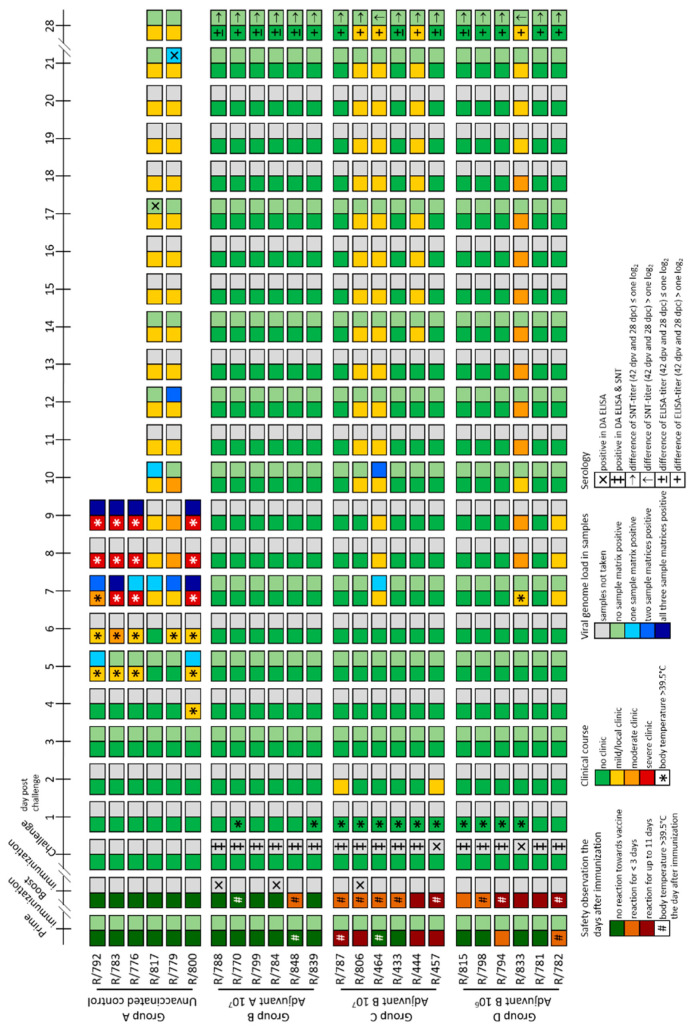
Summarized presentation of adverse reactions after immunization, clinical reaction, viral genome load in different samples matrices and serological examination of the vaccine-efficacy study. A rough overview of adverse reactions after immunization is shown. Furthermore, strength of clinical reaction and rise in body temperature after immunization or challenge infection are presented (left squares of each day) for each animal. Additionally, extent of viremia and shedding of viral genome as well as serological data (right squares of each day) following challenge infection are shown for each individual.

**Figure 8 vaccines-09-00004-f008:**
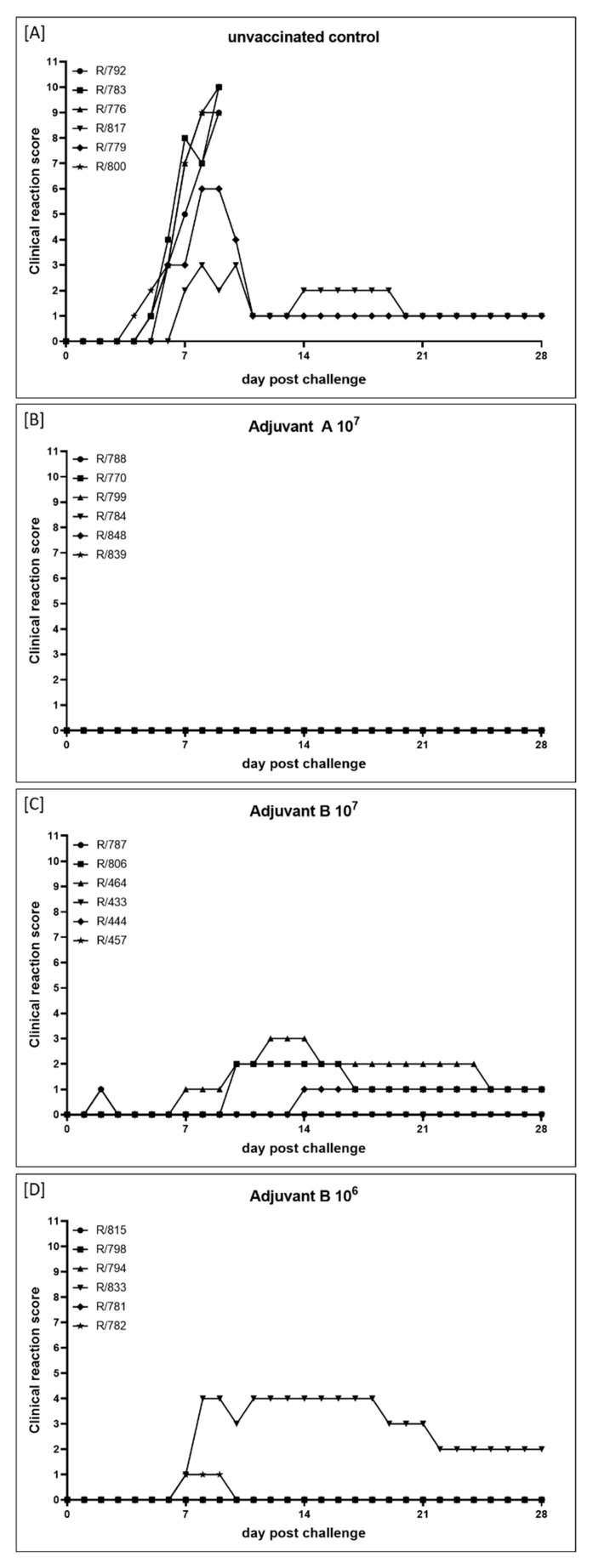
Clinical reaction score of the cattle during the vaccine-efficacy study. (**A**) Cattle of Group 2A serve as mock-control and received PBS. Animals of the other groups were immunized with inactivated LSDV-“Serbia” field strain using different adjuvants and virus titers before inactivation. (**B**) Cattle of Group 2B were vaccinated with Adjuvant A 10^7^, (**C**) cattle of Group 2C received Adjuvant B 10^7^, and (**D**) animals of Group 2D were immunized with Adjuvant B 10^6^ prototype vaccine. Clinical reaction score was calculated daily from the day of challenge infection until 28 dpc.

**Figure 9 vaccines-09-00004-f009:**
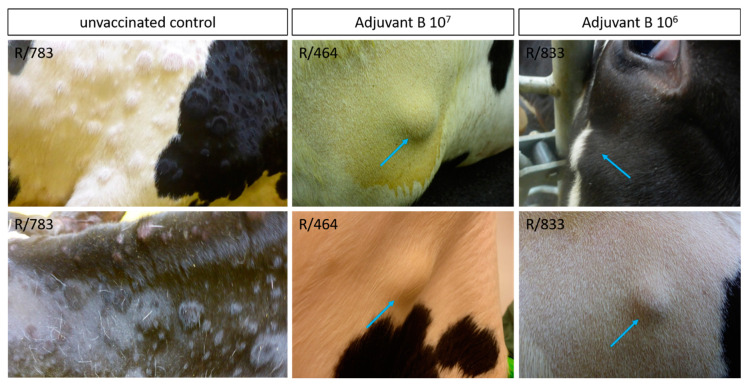
Comparison of skin nodules observed after challenge infection during the vaccine-efficacy study. After challenge infection, skin alterations occurred in some cattle previously immunized with Adjuvant B-containing vaccine. Although affected cattle of the challenge control group showed skin lesions typical for LSDV (e.g., R/783), skin alterations in cattle previously vaccinated with Adjuvant B 10^7^ (e.g., R/464) and Adjuvant B 10^6^ (e.g., R/833), respectively, were clearly different in shape and appearance (blue arrows).

**Figure 10 vaccines-09-00004-f010:**
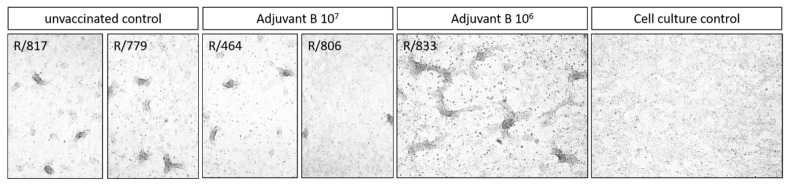
Virus isolation from skin samples obtained from skin nodules of Adjuvant B-vaccinated cattle that were positive for Capripox virus genome. Skin samples were homogenized and incubated on susceptible MDBK cells for 7 days. Afterwards, cytopathic effect was analyzed using a light microscope. Virus isolation was successful of all skin samples that were tested positive for Capripox virus genome.

**Figure 11 vaccines-09-00004-f011:**
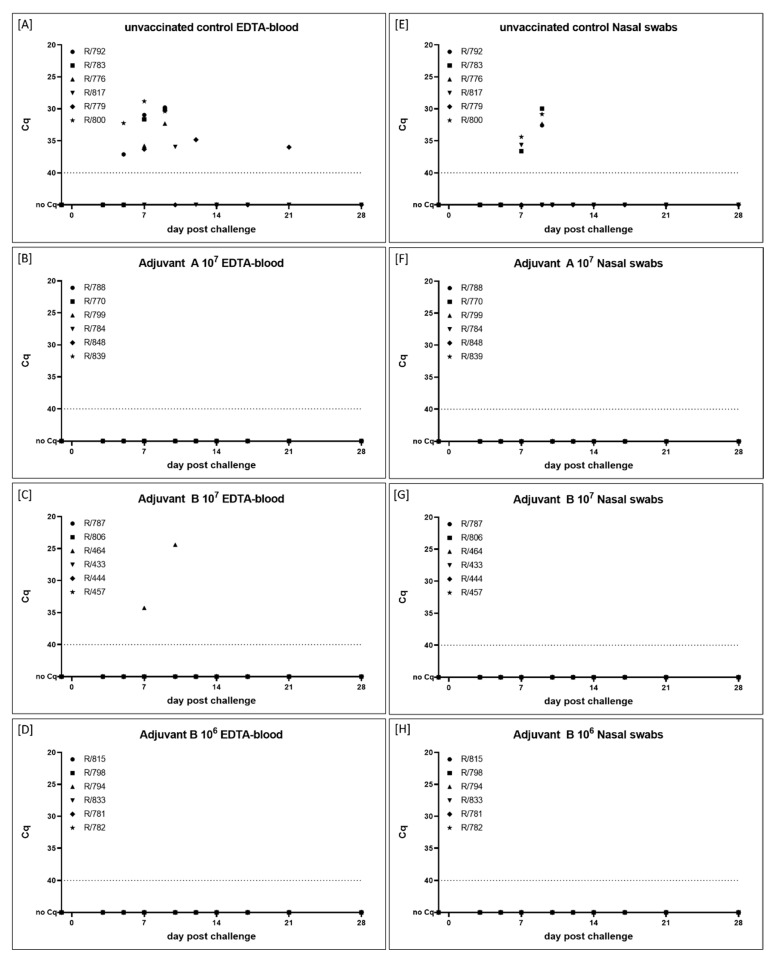
Viral genome load in (**A**–**D**) EDTA-blood and (**E**–**H**) nasal swabs taken during the vaccine-efficacy animal trial. (**A**+**E**) Cattle of Group 2A serve as mock-control and received PBS. Animals of the other groups were immunized with inactivated LSDV-“Serbia” field strain using different adjuvants and virus titers before inactivation. (**B** + **F**) Cattle of Group 2B were vaccinated with Adjuvant A 10^7^, (**C** + **G**) cattle of Group 2C received Adjuvant B 10^7^, and (**D** + **H**) animals of Group 2D were immunized with Adjuvant B 10^6^ prototype vaccine. The samples were taken at certain time points during the animal trial and analyzed regarding the viral genome load. Cut-off was set at Cq 40.0.

**Figure 12 vaccines-09-00004-f012:**
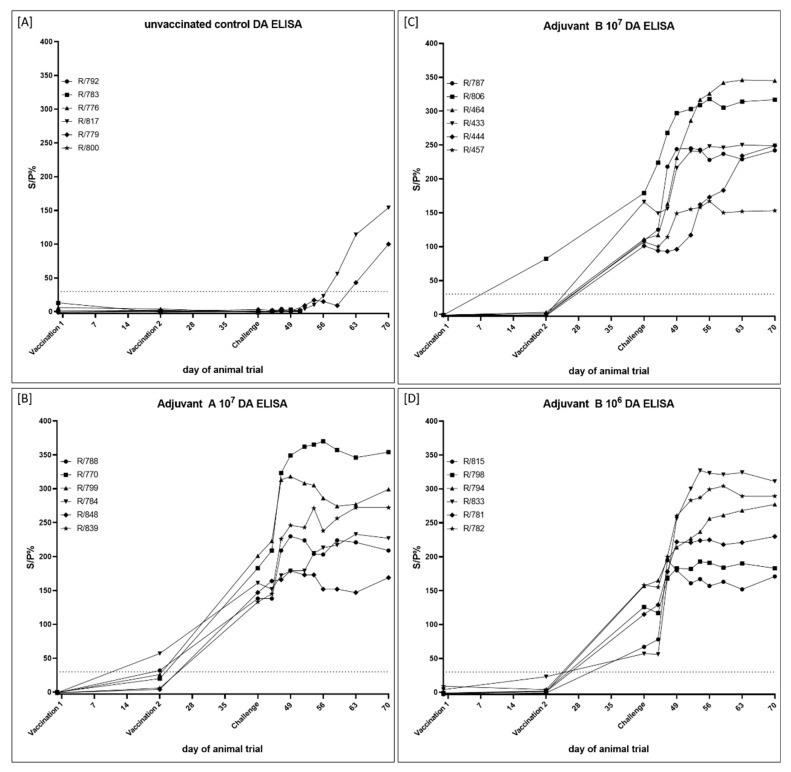
Serological examination of the sera taken during the vaccine-efficacy study using the DA ELISA. (**A**) Cattle of Group 2A serve as mock-control and received PBS. Animals of the other groups were immunized with inactivated LSDV-“Serbia” field strain using different adjuvants and virus titers before inactivation. (**B**) Cattle of Group 2B were vaccinated with Adjuvant A 10^7^, (**C**) cattle of Group 2C received Adjuvant B 10^7^, and (**D**) animals of Group 2D were immunized with Adjuvant B 10^6^ prototype vaccine. Samples were defined positive at an S/P% ratio ≥ 30.

**Table 1 vaccines-09-00004-t001:** Vaccine preparation. TE buffer was used to compensate different volumes of added adjuvants. Dilution of antigen for testing of different amounts was performed using phosphate buffered saline (PBS).

Component	Adjuvant A 10^7^ (Group 2B)	Adjuvant B 10^7^ (Group 2C)	Adjuvant B 10^6^ (Group 2D)
Inactivated antigen	10 mL	10 mL	1 mL
Adjuvant A	2 mL	-	-
Adjuvant B	-	10 mL	10 mL
TE buffer pH 8.0	8 mL	-	-
0.063% PBS	-	-	9 mL

**Table 2 vaccines-09-00004-t002:** Analyses of Capripox virus genome load in samples taken from skin nodes at different time points after challenge infection.

Group	Cattle	Pan-Capripox Real-Time qPCR (Cq)		
		11 dpc	13 dpc	17dpc
Group 2A—unvaccinated control (PBS)	R/779	20.2	n.t.	n.t.
		24.6	n.t.	n.t.
	R/817	21.9	n.t.	n.t.
		20.6	n.t.	n.t.
		23.4	n.t.	n.t.
Group 2B—Adjuvant A 10^7^	R/784	no Cq	n.t.	n.t.
Group 2C—Adjuvant B 10^7^	R/806	21.5	no Cq	n.t.
		n.t.	37.3	n.t.
	R/464	18.1	17.9	n.t.
		n.t.	16.4	n.t.
	R/444	n.t.	n.t.	20.9
		n.t.	n.t.	31.4
Group 2D—Adjuvant B 10^6^	R/833	17.8	18.1	n.t.
		n.t.	17.2	n.t.

n.t. means no sample taken at this certain day.

**Table 3 vaccines-09-00004-t003:** Viral genome load in certain lymph nodes taken during necropsy. Taken samples were analyzed using the pan-Capripox real-time qPCR. Numbers display Cq-values.

Group	Cattle	Cervical Lymph Node	Mediastinal Lymph Node	Mesenterial Lymph Node
Group 2A—unvaccinated control (PBS)	R/792	21.9	27.4	no Cq
	R/783	27.4	23.3	no Cq
	R/776	28.4	35.2	no Cq
	R/817	38.0	no Cq	no Cq
	R/779	no Cq	no Cq	no Cq
	R/800	27.6	35.2	no Cq
Group 2B—Adjuvant A 10^7^	R/788	no Cq	n.a.	n.a.
	R/770	no Cq	n.a.	n.a.
	R/799	no Cq	n.a.	n.a.
	R/784	no Cq	n.a.	n.a.
	R/848	no Cq	n.a.	n.a.
	R/839	no Cq	n.a.	n.a.
Group 2C—Adjuvant B 10^7^	R/787	no Cq	n.a.	n.a.
	R/806	no Cq	n.a.	n.a.
	R/464	no Cq	n.a.	n.a.
	R/433	no Cq	n.a.	n.a.
	R/444	no Cq	n.a.	n.a.
	R/457	no Cq	n.a.	n.a.
Group 2D—Adjuvant B 10^6^	R/815	no Cq	n.a.	n.a.
	R/798	no Cq	n.a.	n.a.
	R/794	no Cq	n.a.	n.a.
	R/833	no Cq	n.a.	n.a.
	R/781	no Cq	n.a.	n.a.
	R/782	no Cq	n.a.	n.a.

n.a. means sample not analyzed.

**Table 4 vaccines-09-00004-t004:** Comparison of the ELISA titer and SNT titer of the sera taken at 42 dpv (≙ 0 dpc) and 28 dpc during the vaccine-efficacy study. → displays difference of titers ≤ one log_2_, ↑ means increase of titer for more than one log_2_, samples with a SNT titer ≥ 1:20 are defined positive.

Group	Cattle	DA ELISA			SNT		
		42 dpv1	28 dpc	Comparison	42 dpv1	28 dpc	Comparison
Group 2B Adjuvant A 10^7^	R/788	1:32	1:64	→	1:50	1:40	→
	R/770	1:32	1:256	↑	1:40	1:16	→
	R/799	1:64	1:128	→	1:80	1:80	→
	R/784	1:16	1:32	→	1:40	1:50	→
	R/848	1:32	1:16	→	1:20	1:13	→
	R/839	1:16	1:64	↑	1:32	1:20	→
Group 2C Adjuvant B 10^7^	R/787	1:16	1:64	↑	1:100	1:80	→
	R/806	1:64	1:256	↑	1:200	1:256	→
	R/464	1:8	1:2048	↑	1:80	1:1024	↑
	R/433	1:32	1:64	→	1:160	1:100	→
	R/444	1:8	1:64	↑	1:128	1:128	→
	R/457	1:8	1:16	→	1:13	1:32	→
Group 2D Adjuvant B 10^6^	R/815	1:8	1:16	→	1:32	1:20	→
	R/798	1:16	1:64	↑	1:64	1:32	→
	R/794	1:32	1:128	↑	1:40	1:50	→
	R/833	1:2	1:1024	↑	1:13	1:400	↑
	R/781	1:8	1:32	↑	1:100	1:32	→
	R/782	1:16	1:128	↑	1:50	1:160	→

## Data Availability

The data presented in this study are available on request from the corresponding author. The data are not publicly available due to partial funding by third party.

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
