# Peer review of "Development of a Safe and Highly Efficient Inactivated Vaccine Candidate against Lumpy Skin Disease Virus"

_vaccines, 2020, doi:10.3390/vaccines9010004_

Round 1

Reviewer 1 Report

This study is of interest so that a safe but effective vaccine can be found. I appreciate the authors attempt to conduct a thorough, in-depth study. However, I found the paper to be unnecessarily dense and hard to get through. For example, the discussion section is much too long and should be condensed. Most of the paper can be condensed. Furthermore, in my opinion, there were methodological flaws and a lack of necessary explanations at many crucial points in the paper.

Some suggestions for wording/grammar improvement:

line 32: globally (the) most

line 42: change "are" to "as" ?

line 45: change "is" to "are"

line 81: change "great" to another word such as "desirable" and strike the word "very"

lines 286 and 574: change "massive" to "severe"

line 641: strike the word "great"

line 648: change "contrarily" to "conversely"

Critique of the Introduction section:

  • I would shorten/condense the current information in this section.
  • This section would benefit from the inclusion of background on the virus itself (i.e., replication cycle, animal reservoirs, method of transmission, etc).

Critique of the Materials and Methods section:

  • Why choose BEI inactivation instead of another method? Explain.
  • If the adjuvant called "A" is commercially available, it's unclear to me why the authors can't actually name it instead of referring to it generically as "A". Is this the actual name?
  • Indicate whether the clinical and safety scoring was done in a blinded fashion or not.
  • It appears that adjuvant B is proprietary yet I see no reference given (section 2.3.1) for how the formulation was arrived at. There must be some published research informing the creation of such an adjuvant. Also, I see no reason why the authors chose a different adjuvant when "A" is apparently in the literature and commercially available. If they want to compare and contrast different adjuvants, then they should set up a proper study to compare the two in a thorough manner (this study was not that). I found the entire inclusion and discussion of different adjuvants to be a distraction to the major goal of the study, which was to compare an attenuated vaccine to an inactivated version due to safety concerns.
  • in section 2.1.2, is there a reference for the pan qPCR process that is mentioned? If not, it needs to be explained in more detail.
  • I did not see an explanation or rationale for why the vaccine and challenge schedule was chosen.
  • How were the vaccine and challenge doses decided upon?
  • Why was such a short interval chosen between vaccination and challenge? This does not seem to mimic a "real world" situation. The long-term immune response and efficacy is of greater interest or significance in my view than a challenge conducted about 2-3 weeks (depending on the trial) after the last vaccine dose.
  • Also related to the vaccine schedule in Figure 1: why is there a different time interval between vaccination and challenge for Herbivac (1B) and the inactivated LSDV group (1C)? The interval is matched (as it should be) in the schedule shown in Figure 2.
  • I did not see any explanation for why the authors chose the virus strains they did for the vaccines AND why they switched from one strain to another between the two vaccine trials. This is a major flaw in my view.

Critique of the Results section:

  • Most of the text in the figures (axis labels, legends, etc.) is hard to read because it's very small.

Author Response

Some suggestions for wording/grammar improvement:

line 32: globally (the) most
line 42: change "are" to "as" ?
line 45: change "is" to "are"
line 81: change "great" to another word such as "desirable" and strike the word "very"
lines 286 and 574: change "massive" to "severe"
line 641: strike the word "great"
line 648: change "contrarily" to "conversely"

We thank the reviewer for the improvement of grammar and wording and corrected this in the manuscript according to the reviewer’s comments.

Critique of the Introduction section:

    I would shorten/condense the current information in this section.

    This sectionwould benefit from the inclusion of background on the virus itself (i.e., replication cycle, animal reservoirs, method of transmission, etc).

Too detailed information were removed and introduction thereby was moderately shortened. Since Reviewer 2 did not comment length and content of the introduction, great changes were not made. In addition, method of transmission and residence of virus during non-vector seasons were added. Detailed description of the replication cycle, however, would enlarge the introduction significantly.

Critique of the Materials and Methods section:

  Why choose BEI inactivation instead of another method? Explain.

BEI is a standard method for inactivation of viruses especially for commercially available inactivated vaccines for animal such as foot-and-mouth disease virus vaccine or bluetongue virus vaccine. This is mentioned in the manuscript now.

If the adjuvant called "A" is commercially available, it's unclear to me why the authors can't actually name it instead of referring to it generically as "A". Is this the actual name?

Name and reference of Adjuvant A were added in the manuscript section 2.2.1. including the adjuvant concentration.

Indicate whether the clinical and safety scoring was done in a blinded fashion or not.

Clinical Scoring as well as safety observation was performed in a non-blinded manner. However, safety scoring was performed by independent employees. These information were included into the manuscript.

It appears that adjuvant B is proprietary yet I see no reference given (section 2.3.1) for how the formulation was arrived at. There must be some published research informing the creation of such an adjuvant. Also, I see no reason why the authors chose a different adjuvant when "A" is apparently in the literature and commercially available. If they want to compare and contrast different adjuvants, then they should set up a proper study to compare the two in a thorough manner (this study was not that). I found the entire inclusion and discussion of different adjuvants to be a distraction to the major goal of the study, which was to compare an attenuated vaccine to an inactivated version due to safety concerns.

Aim of the first study, the proof-of-concept-study, was to analyse the efficacy of an inactivated vaccine against LSDV in cattle in comparison to a commercially available live attenuated vaccine. In the second study, impact of the used adjuvant should be tested in a small scale using an alternative LSDV strain propagated in an established vaccine production cell line. Since the used Adjuvant A is not authorized in Europe, a second adjuvant (Adjuvant B) was included into the vaccine-efficacy-study.

Information of the adjuvants are given by the respective manufacturers. Respective information were added in the revised manuscript.

in section 2.1.2, is there a reference for the pan qPCR process that is mentioned? If not, it needs to be explained in more detail.

We thank the reviewer for this observation. The pan Capripox real-time qPCR used for validation of inactivation is the same assay as used for molecular analyses of the samples taken during the animal trial and described in section 2.1.5 Molecular diagnostics. We refer to it now in the manuscript.

I did not see an explanation or rationale for why the vaccine and challenge schedule was chosen.

How were the vaccine and challenge doses decided upon?

Vaccine doses were chosen due to the recommendation of the manufacturer of the used Herbivac vaccine. In the Herbivac data sheet, 2 ml/dose are recommended for cattle. Vaccine dose should be the same volume, which is why 2 ml were also chosen for the inactivated prototype vaccine. Titer of virus before inactivation was taken as high as possible since no information regarding the minimum protective dose are available until now. Since LSDV grows less good on BHK-21 cells than on MDBK cells, in the second trial two different antigen concentrations were tested: 107 CCID50/ml before inactivation (10x concentrated to reproduce the first trial) and 106 CCID50/ml before inactivation (reflecting a titer normally received with V/99 on BHK-21 cells).

Challenge dose in the first study was chosen due to the only results known for LSDV-“Macedonia2016” field strain in cattle (see Möller et al. Archives of Virology volume 164, pages2931–2941(2019)). In the second study, challenge dose was chosen due to the results of our minimum infective dose-study (see Wolff et al. 2020 Viruses 2020, 12, 768; doi:10.3390/v12070768).

This is now clarified in the manuscript. 

Why was such a short interval chosen between vaccination and challenge? This does not seem to mimic a "real world" situation. The long-term immune response and efficacy is of greater interest or significance in my view than a challenge conducted about 2-3 weeks (depending on the trial) after the last vaccine dose.

We fully agree with the reviewer, that long-term immunity plays an important role during natural outbreaks, and long-lasting studies would be very interesting and should necessarily be performed in the future. However, aim of the presented studies was to test if inactivated LSDV prototype vaccines are generally able to induce immunity in cattle and to determine important factors for safety and efficacy. The relative short interval between vaccination and challenge based mainly on the recommendations of the supplier of different live-attenuated LSDV vaccines. Also here the protection three weeks after vaccination are postulated. As already discussed in the manuscript, minimum protective dose as well as duration of long term immunity have to be studied necessarily.

Also related to the vaccine schedule in Figure 1: why is there a different time interval between vaccination and challenge for Herbivac (1B) and the inactivated LSDV group (1C)?The interval is matched (as it should be) in the schedule shown in Figure 2.

Time schedules were chosen as followed:

Proof-of-concept study: Time schedule for Group B was chosen due to the manufacturer’s information that cattle vaccinated with Herbivac LS are complete protected 3 weeks post vaccination. For the inactivated vaccine group, suboptimal vaccine conditions should be simulated in order to gain better insight in protectivity and efficacy of the inactivated vaccine prototype, which is why challenge infection was performed only 2 weeks after secondary immunization.

Vaccine-efficacy study: Vaccination was performed i.m. at 0 dpv and 21 dpv as it is standardly performed for inactivated vaccines.

This was added to the respective sections of the manuscript.

I did not see any explanation for why the authors chose the virus strains they did for the vaccines AND why they switched from one strain to another between the two vaccine trials. This is a major flaw in my view.

LSDV-“Neethling vaccine” strain was chosen as it is closely related to the live attenuated vaccine strains used for commercial vaccines. In this first proof-of-concept study, impact of different virus isolates should be included to gain insight whether an inactivated vaccine against LSDV is efficient or not.

For development of a safe and efficient inactivated vaccine that can be produced in large scale and to bypass the problems resulting of bovine vaccines produced on bovine cell lines [55], LSDV had to be adapted on a non-bovine cell line. For commercial use, the guaranteed origin and the passage history of the used LSDV isolate should also be traceable documented. These criteria were fulfilled by the LSDV-“Serbia” field strain. These information were added to the respective material and method sections.

In addition, for a commercial vaccine the robust and safe inactivation of the used virus strain must be ensured. Thus, the use of a virulent field strains for the generation of an inactivated vaccine is generally accepted (e.g. FMDV, BTV, SBV). 

Critique of the Results section:

Most of the text in the figures (axis labels, legends, etc.) is hard to read because it's very small.

This is an understandable comment. We thought about showing only median/average, but this is kind of difficult for LSDV. Even in the control group, some animals show severe viremia and viral shedding whereas other show only slight presence or complete absence of viral genome load. Similar pattern can be observed for body temperature, clinical reaction score and seroconversion. Due to these observations, we see no real added value in changing the type of graphical representation. However, we split most of the figures so that cell-associated viremia and viral shedding via nasal swab samples are now presented in the main manuscript and added cell-free viremia as well as results of oral and ocular swab samples in the supplemental material. The same was done with clinical reaction: Body temperature can now be found in the supplemental material and CRS is shown in the main manuscript. This allows enlarged graphics. Summarized figures were not changed, as this scheme was recently approved in another publication (see Wolff et al. Viruses. 2020 Jul; 12(7): 768).

Reviewer 2 Report

Wolff et al. have submitted an interesting report on the ability of inactivated Lumpy Skin Disease virus (LSDV) to vaccinate cattle against challenge with a virulent LSDV stain. The ability to vaccinate with inactivated viruses remains controversial in the Poxvirus field therefore, even though previous studies suggest this is possible, the manuscript by Wolff et al strongly reinforces such findings. The study has been carefully conducted, examining vaccine reactivity, pathology after challenge and the immune response. Several aspects of the manuscript would however need clarification and in some instances improvements.

General comments

Use of an inactivated “field strain” as a vaccine may raise safety issues if they were to be produced in countries that are no longer endemic for LSDV or even countries that remain endemic. Would the authors care to very briefly point this out in their manuscript and maybe justify the choice of a field strain instead of a vaccine strain as a starting point for an inactivated virus vaccine.

The authors mention that they used a standard protocol for virus inactivation with BEI. They should either provide a reference for the protocol or describe it in the methods section.

The vaccine preparation is a crude extract of cell debris and virus. Whether such a preparation actually requires adjuvant for its ability to vaccinate cattle remains unknown as the authors did not perform any experiment without adjuvant. I understand that such experiments are costly and involve a great deal of work but it’s a pity that there’s no information as to whether an adjuvant is indeed required.

More importantly the authors refer to adjuvant A and adjuvant  B instead of providing the exact name, composition and dose of adjuvant used. Scientific publications are meant to inform the community of interesting discoveries that other scientists may wish to further evaluate. Therefore the authors must provide the precise name of the commercial adjuvant used as well as the dosing employed. They should also provide the precise composition and dosing of their in house adjuvant. They are of course free to patent or trademark any adjuvants or vaccination procedures to protect their commercial rights.

The figures are sometimes impossible to read on a printout version of the manuscript because of the small print chosen (figure 3, 4, 5, 6, 7, 8, 11, 12). One solution to this problem would be to provide figures representing mean values of all animals instead of individual animals. Mean values would also give a more global view of the results. The data for individual animals could be provided as a supplementary data if this is accepted policy of the journal. Furthermore, the results section goes into a lot of detail about variations of the various parameters followed after challenge. This description could be somewhat shortened so as to make the major points the authors wish to convey.     

Specific comments and suggestions  

Line 42 change “are” to “as”

Line 42 I suggest changing “Regaining freedom” to “Eradicating the disease”

Line 79 change “can also be not excluded” to “can also not be excluded”

Line 84 I suggest changing “received” to “obtained”

Line 94 change “those” to “that”

Line 111-112 I suggest “In addition to protection against challenge infection, safety and immunogenicity of the inactivated vaccine were analyzed”

Line 310 change “extend” to extent”

Line 543 remove “first”

Round 2

Reviewer 1 Report

I think the authors have made a decent attempt at addressing my concerns.